# Local-Global Associative Frames for Symmetry-Preserving Crystal Structure Modeling

**Haowei Hua**[1], **Wanyu Lin**[1,2*]

[1]Department of Computing, [2]Department of Data Science and Artificial Intelligence
The Hong Kong Polytechnic University, Hong Kong SAR, China
haowei.hua@connect.polyu.hk, wan-yu.lin@polyu.edu.hk

## Abstract

Crystal structures are defined by the periodic arrangement of atoms in 3D space, inherently making them equivariant to SO(3) group. A fundamental requirement for crystal property prediction is that the model's output should remain invariant to arbitrary rotational transformations of the input structure. One promising strategy to achieve this invariance is to align the given crystal structure into a canonical orientation with appropriately computed rotations, or called *frames*. However, existing work either only considers a global frame or solely relies on more advanced local frames based on atoms' local structure. A global frame is too coarse to capture the local structure heterogeneity of the crystal, while local frames may inadvertently disrupt crystal symmetry, limiting their expressivity. In this work, we revisit the frame design problem for crystalline materials and propose a novel approach to construct expressive **S**ymmetry-**P**reserving **Frame**s, dubbed as **SPFrame**, for modeling crystal structures. Specifically, this local-global associative frame constructs invariant local frames rather than equivariant ones, thereby preserving the symmetry of the crystal. In parallel, it integrates global structural information to construct an equivariant global frame to enforce SO(3) invariance. Extensive experimental results demonstrate that SPFrame consistently outperforms traditional frame construction techniques and existing crystal property prediction baselines across multiple benchmark tasks.

## 1 Introduction

Fast and accurate prediction of crystal properties is essential for accelerating the discovery of novel materials, as it enables efficient screening of promising candidates from vast material space [38]. Traditional approaches based on high-fidelity quantum-mechanics calculations, such as density functional theory (DFT), can provide acceptable error margin for property predictions but they require high computational resources [58], thereby limiting their practical deployment. As an alternative, machine learning models have shown great potential for predicting crystal material properties with both efficiency and accuracy. These methods typically leverage 3D geometric graph representations of crystals in conjunction with geometric graph neural networks (GGNNs) [2, 36, 3, 55] or transformer-based variants of GGNNs [57, 49, 56, 27, 51, 19] to establish mappings between crystal structural data and their properties.

When establishing the structure-property mapping, crystal structures are defined by a periodic arrangement of atoms in 3D space, which inherently makes them equivariant under SO(3) group transformations (i.e., rotations). However, many crystal properties, such as formation energy, are scalar quantities that remain invariant under such transformations. Consequently, to ensure accurate property prediction, GGNNs must maintain invariance when input structures are subjected to SO(3)

---

[*]Corresponding author.

39th Conference on Neural Information Processing Systems (NeurIPS 2025).

transformations. For this purpose, early studies have employed SO(3)-invariant features, such as simple interatomic distances [55, 56], which prevent the designed GGNNs from capturing rich interatomic directional information. More recent works has attempted to incorporate interatomic directional information [57] while preserving SO(3)-invariance by carefully designing the network architecture, such as converting directional vectors into angle-based representations [57]. Although this approach successfully ensures invariance, it imposes architectural constraints that can limit the flexibility and scalability of neural networks in modeling complex crystal structures.

Alternatively, the global frame approach can be integrated with any GGNNs without imposing constraints on the network architecture, while still ensuring compliance with the SO(3) invariance requirement. These global frames are constructed in an equivariant manner (such as PCA frame [9]) with respect to the input structure, effectively aligning the structure to a canonical orientation [19, 15, 50]. However, because the same frame is applied to all atoms in the structure, the global frame approach lacks the ability to capture the local structure heterogeneity, limiting their expressivity. To address this limitation, recent developments have shifted toward local frame strategies, where distinct frames are dynamically constructed for each atom based on its local structure [19, 34, 50]. This approach allows for greater differentiation among atomic local environment, thereby improving the expressivity and enhancing the model performance [34, 50]. Despite these advantages, directly applying this general equivariant local frame to crystal structures may unintentionally disrupt the symmetry of the crystal, as discussed in Section 3.1. This disruption hinders the model's ability to distinguish atoms located at Wyckoff positions, thereby weakening its capacity to capture structural details.

To address these challenges, we revisit the problem of frame design for crystals and analyze the root cause of the symmetry breaking observed in previous equivariant local frame methods. Specifically, we identify that constructing local frames based solely on the local atomic structure often breaks the symmetry of the crystal. Motivated by this insight and the symmetry characteristics of crystal structures, we propose a Symmetry-Preserving Frame (SPFrame) method for property prediction. SPFrame constructs invariant local frames rather than equivariant ones. For atoms located at symmetry-equivalent positions, SPFrame assigns identical invariant local frames, which allows their relative local relationships to be preserved after frame transformation. Based on these invariant local frames, an equivariant global frame is further incorporated. Since the same global frame is applied to all atoms, it enforces SO(3) invariance across the structure without disrupting the symmetry of the crystal structure. This local-global associative design enables SPFrame to overcome the symmetry-breaking issue observed in previous local frame methods, enhancing the model's ability to differentiate between distinct atomic local structures. The effectiveness of the SPFrame approach is evaluated on two widely used benchmark datasets for crystalline materials.

## 2 Preliminaries

### 2.1 Coordinate Systems for Crystal Structure Representation

Crystalline materials are defined by a periodic 3D arrangement of atoms, where the smallest repeating unit, known as the unit cell, fully determines the entire crystal structure. Prior studies [56, 52, 21] have established two primary coordinate systems for representing such crystal structure mathematically.

**Cartesian Coordinate System.** A crystal structure is formally defined by the triplet $\mathbf{M} = (\mathbf{A}, \mathbf{X}, \mathbf{L})$. The matrix $\mathbf{A} = [\boldsymbol{a}_1, \boldsymbol{a}_2, \cdots, \boldsymbol{a}_n]^\top \in \mathbb{R}^{n \times d_a}$ contains feature vectors for $n$ atoms within a unit cell, where each row $\boldsymbol{a}_i \in \mathbb{R}^{d_a}$ describes the individual atom feature. The 3D Cartesian coordinates of $n$ atoms in the unit cell are encoded in $\mathbf{X} = [\boldsymbol{x}_1, \boldsymbol{x}_2, \cdots, \boldsymbol{x}_n]^\top \in \mathbb{R}^{n \times 3}$. The lattice maxtrix $\mathbf{L} = [\boldsymbol{l}_1, \boldsymbol{l}_2, \boldsymbol{l}_3] \in \mathbb{R}^{3 \times 3}$ consists of the lattice vectors $\boldsymbol{l}_1, \boldsymbol{l}_2$, and $\boldsymbol{l}_3$, which form the basis of the 3D space. The complete crystal structure emerges through periodic repetition: $(\hat{\mathbf{A}}, \hat{\mathbf{X}}) = \{(\hat{\boldsymbol{a}}_i, \hat{\boldsymbol{x}}_i) | \hat{\boldsymbol{x}}_i = \boldsymbol{x}_i + k_1 \boldsymbol{l}_1 + k_2 \boldsymbol{l}_2 + k_3 \boldsymbol{l}_3, \hat{\boldsymbol{a}}_i = \boldsymbol{a}_i, k_1, k_2, k_3 \in \mathbb{Z}, i \in \mathbb{Z}, 1 \leq i \leq n\}$, where integer coefficients $k_1, k_2, k_3$ generate all possible atomic positions in the periodic lattice.

**Fractional Coordinate System.** This system employs lattice vectors $\boldsymbol{l}_1, \boldsymbol{l}_2$, and $\boldsymbol{l}_3$ as basis, with atomic positions expressed as $\boldsymbol{f}_i = [f_{i,1}, f_{i,2}, f_{i,3}]^\top \in [0, 1)^3$. The conversion to Cartesian coordinates is defined as $\boldsymbol{x}_i = \sum_j f_{i,j} \boldsymbol{l}_j$, where $j = 1, 2, 3$. This yields an crystal representation $\mathbf{M} = (\mathbf{A}, \mathbf{F}, \mathbf{L})$, where $\mathbf{F} = [\boldsymbol{f}_1, \cdots, \boldsymbol{f}_n]^\top \in [0, 1)^{n \times 3}$ contains the fractional coordinates of all atoms in the unit cell.

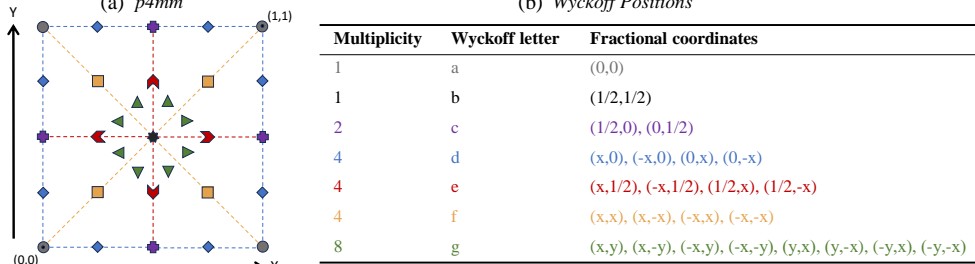

Figure 1: An illustration with 2D plain group P4mm [52, 14, 21]. (a) The figure illustrates the lattice of the P4mm space group, visually demonstrating equivalent positions; the symmetry-equivalent positions are indicated by the same color. (b) The table depicts the Wyckoff positions present in this lattice.

## 2.2 The Symmetry of Crystal Structures

**SO(3) group.** The SO(3) group consists of all rotations in 3D space. Its elements are rotation matrices defined as $\{\mathbf{Q} \mid \mathbf{Q} \in \mathbb{R}^{3\times3}, \mathbf{Q}^\top \mathbf{Q} = \mathbf{I}, \det(\mathbf{Q}) = 1\}$. When applied to crystal data $\mathbf{M}$, an SO(3) transformation yields $\mathbf{M}' = (\mathbf{A}, \mathbf{Q}\mathbf{X}, \mathbf{Q}\mathbf{L})$.

**Space group.** The E(3) group encompasses all rigid transformations including rotations, reflections, and translations. Its elements can be denoted by the pair $\{(\mathbf{Q}, \mathbf{t}) \mid \mathbf{Q} \in \mathbb{R}^{3\times3}, \mathbf{Q}^\top \mathbf{Q} = \mathbf{I}, \mathbf{t} \in \mathbb{R}^3\}$, where $\mathbf{Q}$ is an orthogonal matrix and $\mathbf{t}$ is a translation vector. When an E(3) transformation is applied to the crystal data $\mathbf{M}$, certain elements $(\mathbf{Q}_g, \mathbf{t}_g)$ can map a crystal structure back onto itself due to the inherent symmetry of the structure. These specific elements $(\mathbf{Q}_g, \mathbf{t}_g)$ are collectively referred to space groups. Mathematically, $(\mathbf{A}, \mathbf{Q}_g\mathbf{X} + \mathbf{t}_g, \mathbf{Q}_g\mathbf{L}) = (\mathbf{A}, \mathbf{X}, \mathbf{L})$, where the symbol '=' indicates the equivalence between geometric structures.

**Wyckoff positions.** The concept of space groups leads to the definition of Wyckoff positions, which are sets of symmetry-equivalent atomic sites within a unit cell [20, 21]. Each Wyckoff position is characterized by three fundamental attributes: multiplicity, Wyckoff letter, and fractional coordinates. As shown in Fig. 1, for 2D plain group P4mm, the Wyckoff positions obey specific coordinate constraints, including $0 \le x \le 0.5$, $0 \le y \le 0.5$, $x \le y$, and the identical atomic occupation requirement [20, 21].

## 2.3 Crystal Structure Invariant Learning

**SO(3)-invariance requirement for crystal properties prediction.** The SO(3) group transformation can alter the orientation of a crystal structure within 3D space [57]. Nevertheless, many critical material properties, such as formation energy, are invariant under the SO(3) group transformation. Consequently, for effective crystal property prediction, it is essential that the model can exhibit SO(3)-invariant prediction capabilities. Specifically, for a prediction model denoted as $f_\theta(\cdot)$, if it is SO(3)-invariant, for any rotation matrix $\mathbf{Q} \in \mathbb{R}^{3\times3}$, the following equality holds:

$$f_\theta(\mathbf{A}, \mathbf{Q}\mathbf{X}, \mathbf{Q}\mathbf{L}) = f_\theta(\mathbf{A}, \mathbf{X}, \mathbf{L}). \tag{1}$$

**Frame.** Frame-based methodologies have shown promising in enforcing equivariance and invariance in geometric deep learning [50, 33, 37]. In the context of SO(3) group transformations, a frame can be interpreted as a rotation matrix $\mathbf{F} \in \mathbb{R}^{3\times3}$, deriving from a SO(3)-equivariant map denoted as $h(\mathbf{X})$. This frame transforms the atomic positions $\mathbf{X}$ into an SO(3)-invariant representation represented as $\mathbf{X}\mathbf{F}^\top$. Crucially, this representation remains unchanged under arbitrary rotations $\mathbf{Q}$: $\mathbf{X}\mathbf{F}^\top \xrightarrow{\mathbf{Q}} \mathbf{X}\mathbf{Q}(\mathbf{F}\mathbf{Q})^\top = \mathbf{X}\mathbf{Q}\mathbf{Q}^\top\mathbf{F}^\top = \mathbf{X}\mathbf{F}^\top$, thus decoupling the SO(3) invariance requirement for neural network design. Additionally, the concept of a global frame involves using a single frame for all atoms within the atomic system, whereas a local frame is defined for each atom individually, with its calculation based on the atom's local structure. A more detailed discussion of related works can be found in Section 4.

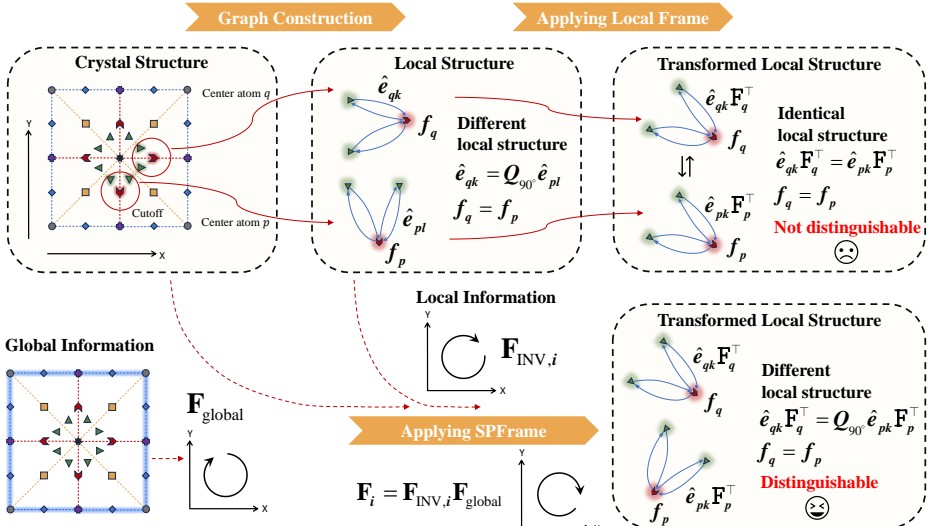

Figure 2: Using the 2D plane group P4mm as a running example, we demonstrate why local frame methods may disrupt the symmetry of crystal. For atoms $p$ and $q$ that belong to the same Wyckoff position type, their local structures can be related by a 90-degree rotation after graph construction. Since equivariant local frames are constructed solely based on local structural information, the resulting frames for $p$ and $q$ also exhibit a 90-degree rotational relationship, applying these frames eliminates the relative orientation between the two local structures. In contrast, SPFrame preserves these relative structural differences by incorporating global structural information during frame construction.

## 3 Methodology

In this section, we first outline the motivation behind our work, with a particular emphasis on the limitations of existing local frame methods when applied to crystal structures, as discussed in Section 3.1. To address these challenges, we introduce the proposed SPFrame method, a local-global associative frame. We further incorporate this method into the established crystal property prediction architecture, yielding a new framework for crystal structure modeling, as described in Section 3.2.

### 3.1 Symmetry Breaking Induced by the Local Frame

Building upon previous works [22, 57, 51], we begin by presenting the general formulation of message passing at $k$-th layer in GGNNs using SO(3)-equivariant edge features, defined as follows:

$$\boldsymbol{f}_i^{(k)} = \psi^{(k)} \left( \boldsymbol{f}_i^{(k-1)}, \sum_{j \in \mathcal{N}(i)} \phi^{(k)} \left( \boldsymbol{f}_i^{(k-1)}, \boldsymbol{f}_j^{(k-1)}, \boldsymbol{e}_{ij}, \widehat{\boldsymbol{e}}_{ij} \right) \right), \tag{2}$$

where $\boldsymbol{f}_i^{(k)}$ denotes the feature vector of atom $i$, $\boldsymbol{e}_{ij} \in \mathbb{R}^d$ represents SO(3)-invariant edge features (e.g., embeddings of interatomic distances between atoms $i$ and $j$), and $\widehat{\boldsymbol{e}}_{ij} \in \mathbb{R}^3$ corresponds to SO(3)-equivariant edge features capturing directional information between atoms (e.g., the edge vector between atoms $i$ and $j$). The functions $\phi^{(k)}(\cdot)$ and $\psi^{(k)}(\cdot)$ are learnable non-linear mappings that define the message construction and aggregation processes, respectively.

When integrating the local frame into the message passing framework, the local frame is adaptively defined for each atom, thereby transforming the equivariant edge features into invariant features. Consequently, the message passing process is reformulated as follows:

$$\boldsymbol{f}_i^{(k)} = \psi^{(k)} \left( \boldsymbol{f}_i^{(k-1)}, \sum_{j \in \mathcal{N}(i)} \phi^{(k)} \left( \boldsymbol{f}_i^{(k-1)}, \boldsymbol{f}_j^{(k-1)}, \boldsymbol{e}_{ij}, \widehat{\boldsymbol{e}}_{ij} \mathbf{F}_i^\top \right) \right), \tag{3}$$

where $\mathbf{F}_i$ denotes the local frame for atom $i$. As illustrated in Section 2.3, the presence of local frames ensures that the output of the GGNNs remains invariant under the SO(3) group transformation.

However, as discussed in Section 2.2, the symmetry of crystal structures implies that atoms occupying the same Wyckoff position type exhibit similar local structures. As illustrated in Figure 2, when constructing the crystal graph, atoms $p$ and $q$, which belong to the same Wyckoff position type, share identical atom features and invariant edge features (such as interatomic distances). The only distinction between these atoms lies in equivariant edge features. These equivariant features, representing the relative directional vectors of atoms $p$ and $q$ with respect to their neighboring atoms, are related through a 90 degrees rotation matrix $\mathbf{Q}_{90^\circ}$.

When local frames are incorporated into GGNNs, they serve to canonicalize the equivariant edge features $\widehat{e}_{ij}$ by mapping them to invariant representations $\widehat{e}_{ij}\mathbf{F}_i^\top$. Since the local frames $\mathbf{F}_p$ and $\mathbf{F}_q$ are constructed equivariantly based on the local structures, it follows that $\mathbf{F}_q = \mathbf{Q}_{90^\circ}\mathbf{F}_p$. As a result, atoms $p$ and $q$, which initially exhibit distinct orientations in their respective local structures, are aligned to a common orientation, rendering their local structures indistinguishable. This process diminishes the model's ability to distinguish symmetry-equivalent yet spatially distinct atoms, thereby limiting the expressivity of GGNNs. Furthermore, such equivalence between atoms $p$ and $q$ under SO(3) transformations is commonly observed in crystals with screw axes or rotational symmetries, such as those found in space groups like P2$_1$, among others [14].

### 3.2 Our SPFrame

To address the challenges outlined above, several critical considerations must be taken into account. First, it is essential to decouple the SO(3) invariance requirement imposed on GGNNs when employing local frames. Simultaneously, for atoms located at equivalent Wyckoff positions, it is imperative to preserve the relative relationships within their local structures following the application of the local frame. This preservation ensures that the GGNN can effectively differentiate between these atoms. Second, in line with the standard definition of local frames, distinct frames should be assigned to atoms occupying non-equivalent positions.

**SO(3) symmetry decoupling and crystal symmetry preserving.** As discussed in Section 3.1, conventional local frame construction assigns different frames to symmetry-equivalent atoms $p$ and $q$, which can inadvertently disrupt the crystal's symmetry. To mitigate this issue in Figure 2, a straightforward yet effective strategy is to assign identical frames to atoms $p$ and $q$. This design ensures that the relative orientation relationships between their local structures are preserved after the frame transformation. For atoms that are not symmetry-equivalent, distinct frames should be constructed to reflect the differences in their local structures. Therefore, We now introduce the SPFrame, defined as

$$\mathbf{F}_i = \mathbf{F}_{\text{INV},i}\mathbf{F}_{\text{global}}, \tag{4}$$

where $\mathbf{F}_{\text{INV},i}$ denotes the invariant local frame for atom $i$ and $\mathbf{F}_{\text{global}}$ denotes the equivariant global frame shared across the atomic system. Using SPFrame, we reformulate Equation 3 as follows:

$$\boldsymbol{f}_i^{(k)} = \psi^{(k)}\left(\boldsymbol{f}_i^{(k-1)}, \sum_{j\in\mathcal{N}(i)}\phi^{(k)}\left(\boldsymbol{f}_i^{(k-1)}, \boldsymbol{f}_j^{(k-1)}, \boldsymbol{e}_{ij}, \widehat{\boldsymbol{e}}_{ij}\mathbf{F}_{\text{global}}^\top\mathbf{F}_{\text{INV},i}^\top\right)\right). \tag{5}$$

Under an SO(3) group transformation applied to the entire atomic system, the presence of global frames ensures that the output of the GGNNs remains invariant, thereby decoupling the SO(3) invariance requirement. Since the global frame is computed based on global structural information, it remains consistent across symmetry-equivalent atoms and thus does not disrupt the symmetry of the crystal. At the same time, the invariant local frame $\mathbf{F}_{\text{INV},i}$ is constructed using an invariant method based on local structural information. Thus, atoms $p$ and $q$, which occupy the same type of Wyckoff position, are assigned identical local frames. Consequently, the transformed SO(3)-invariant representation, $\widehat{e}_{ij}\mathbf{F}_{\text{global}}^\top\mathbf{F}_{\text{INV},i}^\top$, remains distinguishable for atoms $p$ and $q$, while preserving their relative structural differences.

This local-global associative design enables the model to satisfy SO(3) invariance while maintaining the crystal's symmetry. Furthermore, since the invariant local frame $\mathbf{F}_{\text{INV},i}$ still considers local information, the SPFrame for non-symmetry-equivalent positions are computed differently. In addition, we provide a theoretical justification for the superiority of SPFrame over the local frame, as detailed in Appendix A.1. This work also guarantees SE(3) invariance, as detailed in Appendix A.2.

---

**Algorithm 1** Quaternion to Rotation Matrix Conversion

---

**Require:** Quaternion $\boldsymbol{q} = [a, b, c, d] \in \mathbb{R}^4$

**Ensure:** Rotation matrix $\mathbf{Q} \in \mathbb{R}^{3 \times 3}$

1: Normalize the quaternion:

$$s = \sqrt{a^2 + b^2 + c^2 + d^2}, \quad a \leftarrow a/s, \quad b \leftarrow b/s, \quad c \leftarrow c/s, \quad d \leftarrow d/s$$

2: Compute the rotation matrix $\mathbf{Q}$:

$$\mathbf{Q} = \begin{bmatrix} a^2 + b^2 - c^2 - d^2 & 2(bc - ad) & 2(bd + ac) \\ 2(bc + ad) & a^2 - b^2 + c^2 - d^2 & 2(cd - ab) \\ 2(bd - ac) & 2(cd + ab) & a^2 - b^2 - c^2 + d^2 \end{bmatrix}$$

---

**Symmetry-preserving frame construction.** As illustrated in Equation 4, the proposed SPFrame consists of two components: a shared global frame $\mathbf{F}_{\text{global}}$ applied to all atoms, and a set of atom-specific invariant local frames $\mathbf{F}_{\text{INV},i}$. The global frame $\mathbf{F}_{\text{global}}$ plays a key role in decoupling the SO(3) invariance requirement from the GGNN. For simplicity, we propose to use non-parametric approaches such as QR decomposition [33]. Additional implementation details can be found in Appendix A.3.

Correspondingly, the invariant local frame $\mathbf{F}_{\text{INV},i}$ is designed to enhance the expressive power of the GGNN. Since it does not need to independently enforce SO(3) symmetry constraints, this component allows for better flexibility in frame construction. To this end, we employ quaternions [42, 46, 13] as a compact and numerically stable representation of the rotations.

Specifically, we first predict a quaternion for each atom based on its local structure. Inspired by previous work [50, 34], we leverage a message passing scheme to generate the quaternion embeddings:

$$\boldsymbol{q_i} = \psi \left( \boldsymbol{f}_i, \sum_{j \in \mathcal{N}(i)} \phi \left( \boldsymbol{f}_i, \boldsymbol{f}_j, \boldsymbol{e}_{ij} \right) \right), \quad \mathbf{F}_{\text{INV},i} = \text{LF}(\boldsymbol{q_i}), \tag{6}$$

where $\boldsymbol{q}_i \in \mathbb{R}^4$ denotes the predicted quaternion for atom $i$. The message function $\phi(\cdot)$ and the aggregation function $\psi(\cdot)$ can be instantiated using any SO(3)-invariant message passing architecture. In this work, we adopt transformer-based implementations following Yan et al. [56]. Once the quaternion $\boldsymbol{q}_i$ is obtained, it is converted into the corresponding rotation matrix [42] via the mapping LF($\cdot$) (Further details can be found in Appendix A.4). The pseudocode for LF($\cdot$) is presented in Algorithm 1.

**Network architecture.** As demonstrated in previous studies [50, 34], local frames can be effectively integrated into GGNNs that utilize equivariant edge features for message passing. Among these models, eComFormer [57] represents the state of the art in crystal property prediction, leveraging equivariant edge features to enhance message propagation. Based on this, we adopt eComFormer as the backbone architecture for implementing and evaluating the proposed SPFrame strategy. More details on the integration of SPFrame into eComFormer can be found in Appendix A.5.

## 4  Related Works

**Crystal Property Prediction.** GGNNs and their transformer-based extensions (hereafter collectively referred to as GGNNs for convenience) have been widely adopted in crystal property prediction due to their capacity to model complex atomic interactions. Several representative methods, such as CGCNN [55], MEGNet [2], GATGNN [36], Matformer [56], PotNet [32], DOSTransformer [27], and CrystalFormer [49], construct crystal graphs by utilizing SO(3)-invariant interatomic distances as edge features. By avoiding the use of equivariant directional vectors, these models ensure SO(3)-invariant predictions. Similarly, models such as ALIGNN [3], M3GNet [1], Crystalformer [51], and iComFormer [57] leverage invariant angular information as edge features to maintain SO(3)-invariance in prediction. Beyond these, several methods adopt more specialized strategies. For example, eComFormer [57] utilizes equivariant edge features, which subsequently are transformed into two-hop invariant angular representations for preserving invariance.

**Global Frame.** Frames are widely used in both equivariant and invariant learning. However, earlier frame methods, such as the frame averaging (FA) method [41, 9], rely on frame construction techniques like PCA, which produce non-unique frames. This necessitates the use of specially designed loss functions during training to learn invariant representations across all frame-transformed variants. More recently, minimal frame methods [33] have adopted frame construction techniques such as QR decomposition to produce unique frames, thereby improving the efficiency of frames. This is the type of global frame described in this work. It is worth noting that another approach in equivariant and invariant learning, i.e. canonicalization [37, 23, 10, 43], is equivalent to the minimal frame method to some extent [37].

**Local Frame.** Similar to the global frame, the local frame is also a method that transforms equivariant data representations into invariant ones [7, 50, 8, 40, 34, 19]. The difference lies in that the local frame approach generates a separate frame for each atom in the atomic system, which can enhance the expressiveness of GGNNs [50]. Recent work, Crystalframer [19], was the first to introduce local frames into the field of crystal property prediction. Building upon the attention mechanism from [49], it designed two types of equivariant local frames and recalculated different local frames at various network layers. However, as mentioned above, the use of general equivariant local frames may unintentionally decouple the symmetry of the crystal structure.

Beyond the aforementioned studies, we also review approaches that incorporate crystal symmetry into method design, together with other key strategies for enhancing predictive accuracy. A more discussion is provided in Appendix A.6.

## 5    Experiments

To validate the effectiveness of the proposed SPFrame, we performed a comprehensive series of experiments on crystal property prediction. Additionally, we conducted comparative analyses between our method and existing equivariant local frame approaches. A detailed summary of the experimental setup and results is provided below.

### 5.1    Experimental setup

**Datasets.** We utilize two widely used crystal property benchmark datasets: JARVIS-DFT and Materials Project (MP). Following previous work [56, 57, 19], we perform predction tasks of formation energy, total energy, bandgap, and energy above hull (E hull) on JARVIS-DFT dataset. For the MP dataset, we perform predction tasks of formation energy, bandgap, bulk modulus, and shear modulus.

**Baseline Methods.** We selected several state-of-the-art methods in the field, including CGCNN [55], SchNet [45], MEGNet [2], GATGNN [36], ALIGNN [3], Matformer [56], PotNet [32], Crystalformer [49], eComFormer [57], iComFormer [57], and Crystalframer [19], as baseline methods for comparison.

**Frame Comparison and Ablation Studies.** In addition to the aforementioned crystal property prediction methods, we also conducted comparative experiments by replacing the proposed SPFrame with other frame methods integrated into the backbone network. The first method is an SO(3)-equivariant local frame. Inspired by Wang and Zhang [50] and Lippmann et al. [34], we design this approach using Gram-Schmidt orthogonalization to construct SO(3)-equivariant local frames. This method serves as a baseline for examining the impact of breaking the symmetry of crystal structures on model performance. The second method is an SPFrame variant constructed using Gram-Schmidt orthogonalization, allowing for a more direct comparison with the first method, as both rely on the same orthogonalization procedure. This comparison further enables the evaluation of the advantages of the quaternion-based SPFrame. Additional details on the design of these two frame baselines can be found in Appendix A.7.

**Experimental Settings.** Following prior work [57], we evaluate model performance using Mean Absolute Error (MAE) and optimize all models using the Adam optimizer. We conduct our experiments on NVIDIA GeForce RTX 3090 GPUs, with complete hyperparameter configurations (including learning rates, batch sizes, and training epochs) provided in Appendix A.8. In our evaluation, we highlight the best-performing results in bold and indicate second-best performances with underlining.

## 5.2 Experimental Results

**JARVIS.** Table 1 presents the experimental results on JARVIS. The eComFormer architecture, when combined with the our proposed SPFrame, achieves best performance on all prediction tasks, demonstrating consistent improvements over existing approaches. In the comparison of different frame methods, the performance of the SO(3)-equivariant Gram-Schmidt local frame is inferior to that of both the Gram-Schmidt-based SPFrame and the quaternion-based SPFrame in all prediction tasks. This observation confirms that maintaining crystal symmetry while applying local frames enables GGNNs to better distinguish between atoms, leading to improved prediction accuracy. Furthermore, the superior performance of the quaternion-based SPFrame over the Gram-Schmidt-based SPFrame emphasizes that quaternion-derived rotation matrices provide a more effective representation for frame construction in crystal materials. To further demonstrate the generality of SPFrame, we conducted additional experiments combining SPFrame with another backbone architectures. The corresponding results are provided in Appendix A.9.

Table 1: Property prediction results on the JARVIS dataset.

| Method | Form. energy | Total energy | Bandgap (OPT) | Bandgap (MBJ) | E hull |
|---|---|---|---|---|---|
| | eV/atom | eV/atom | eV | eV | eV |
| CGCNN | 0.063 | 0.078 | 0.20 | 0.41 | 0.17 |
| SchNet | 0.045 | 0.047 | 0.19 | 0.43 | 0.14 |
| MEGNet | 0.047 | 0.058 | 0.145 | 0.34 | 0.084 |
| GATGNN | 0.047 | 0.056 | 0.17 | 0.51 | 0.12 |
| ALIGNN | 0.0331 | 0.037 | 0.142 | 0.31 | 0.076 |
| Matformer | 0.0325 | 0.035 | 0.137 | 0.30 | 0.064 |
| PotNet | 0.0294 | 0.032 | 0.127 | 0.27 | 0.055 |
| iComFormer | 0.0272 | 0.0288 | 0.122 | 0.26 | 0.047 |
| Crystalformer | 0.0306 | 0.0320 | 0.128 | 0.30 | 0.046 |
| Crystalframer | 0.0263 | 0.0279 | 0.117 | 0.242 | 0.047 |
| eComFormer | 0.0284 | 0.0315 | 0.124 | 0.283 | 0.044 |
| —w/ SO(3)-equivariant Gram-Schmidt local frame | 0.0285 | 0.0296 | 0.115 | 0.271 | 0.043 |
| **—w/ Gram-Schmidt-based SPFrame (ours)** | 0.0268 | 0.0281 | 0.109 | 0.259 | 0.043 |
| **—w/ Quaternion-based SPFrame (ours)** | **0.0261** | **0.0276** | **0.107** | **0.239** | **0.042** |

**MP.** Table 2 presents the experimental results on the MP dataset. Similar to the results obtained on the JARVIS dataset, the eComFormer architecture, when combined with our proposed SPFrame, achieves the best performance on two out of four prediction tasks. The comparison with different frame methods further demonstrates the effectiveness of SPFrame. Furthermore, considering that the performance of the our method on bulk modulus and shear modulus prediction was not particularly strong on the MP dataset, we additionally conducted experiments on the JARVIS dataset for these two properties. The corresponding results are provided in Appendix A.9.

Table 2: Property prediction results on the MP dataset.

| Method | Formation energy | Bandgap | Bulk modulus | Shear modulus |
|---|---|---|---|---|
| | eV/atom | eV | log(GPa) | log(GPa) |
| CGCNN | 0.031 | 0.292 | 0.047 | 0.077 |
| SchNet | 0.033 | 0.345 | 0.066 | 0.099 |
| MEGNet | 0.030 | 0.307 | 0.060 | 0.099 |
| GATGNN | 0.033 | 0.280 | 0.045 | 0.075 |
| ALIGNN | 0.022 | 0.218 | 0.051 | 0.078 |
| Matformer | 0.021 | 0.211 | 0.043 | 0.073 |
| PotNet | 0.0188 | 0.204 | 0.040 | 0.065 |
| iComFormer | 0.0183 | 0.193 | 0.0380 | **0.0637** |
| Crystalformer | 0.0186 | 0.198 | 0.0377 | 0.0689 |
| Crystalframer | 0.0172 | 0.185 | **0.0338** | 0.0677 |
| eComFormer | 0.0182 | 0.202 | 0.0417 | 0.0729 |
| —w/ SO(3)-equivariant Gram-Schmidt local frame | 0.0183 | 0.187 | 0.0407 | 0.0721 |
| **—w/ Gram-Schmidt-based SPFrame (ours)** | 0.0174 | 0.191 | 0.0370 | 0.0678 |
| **—w/ Quaternion-based SPFrame (ours)** | **0.0171** | **0.181** | 0.0371 | 0.0672 |

**Efficiency comparison.** Table 3 compares the model efficiency of several frame methods and the skeleton method, eComFormer. We show the average training time per epoch, total number of parameters, and average testing time per material, all evaluated on the JARVIS-DFT formation energy dataset. The batch size is kept consistent across all experiments, and all experiments are conducted using a single NVIDIA GeForce RTX 3090 GPU. Due to the presence of trainable network

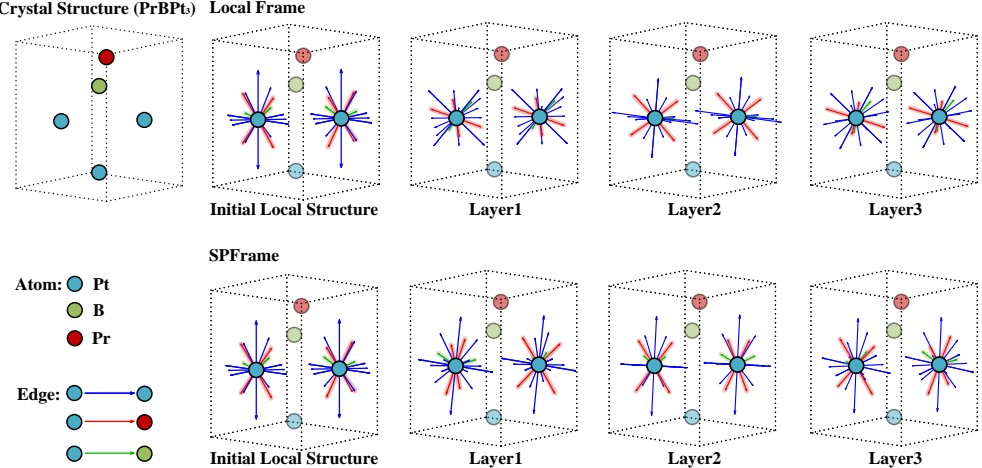

Figure 3: Visual analysis. After graph construction, atoms at symmetry-equivalent positions may exhibit distinct local structures. Local frame methods tend to transform these local structures into identical representations, thereby removing the relative differences and making the atoms indistinguishable to the model. In contrast, SPFrame preserves these structural distinctions, enabling t he model to effectively differentiate between atoms located at symmetry-equivalent positions.

components in the frame calculations, all frame-based methods are less efficient than the skeleton method, eComFormer. The SO(3)-equivariant Gram-Schmidt local frame and Gram-Schmidt-based SPFrame, which construct rotation matrices using the Gram-Schmidt orthogonalization method, require two distinct message passing and aggregation modules to generate and orthogonalize two different vectors (see Appendix A.7). Consequently, their efficiencies are similar but lower than that of SPFrame. In contrast, SPFrame only requires a single message passing and aggregation module to predict quaternions, which are subsequently used to construct the rotation matrix, thereby reducing computational cost.

Table 3: Efficiency analysis.

| Method | Num. Params. | Time/epoch | Test time/Material |
|---|---|---|---|
| eComFormer | 4.9 M | 127.86 s | 31.76 ms |
| —w/ SO(3)-equivariant Gram-Schmidt local frame | 8.5 M | 235.42 s | 43.43 ms |
| —w/ Gram-Schmidt-based SPFrame | 8.5 M | 234.86 s | 42.81 ms |
| —w/ Quaternion-based SPFrame | 6.3 M | 143.75 s | 37.07 ms |

**Visual analysis.** To empirically evaluate the limitations of the equivariant local frame method and the effectiveness of SPFrame, we present a concrete visual example, as shown in Figure 3. Specifically, we visualize the crystal structure of PrBPt$_3$ (JVASP-16632) and illustrate how different frames influence atoms located at symmetry-equivalent positions in the context of the formation energy prediction task on the JARVIS-DFT dataset. For the two symmetry-equivalent Pt atoms, the equivariant local frame transforms the edge features such that their local structures become indistinguishable. In contrast, SPFrame preserves the relative structural differences between these atoms after transformation, enabling the model to distinguish them. On this sample, the backbone network using the equivariant local frame yields a MAE of 0.0551, while the same backbone integrated with SPFrame achieves a lower MAE of 0.0503, demonstrating superior predictive accuracy.

## 6 Conclusion

This paper investigates the limitations of applying conventional local frame methods to crystal structures. Although these local frame methods enable GGNNs to satisfy the SO(3) invariance requirement, they may inadvertently disrupt the symmetry of the crystal, limiting the model's ability to distinguish atoms situated at symmetry-equivalent positions. To address this challenge, we propose the SPFrame for crystal property prediction. SPFrame constructs frames by incorporating both local

atomic structural information and global structural information. Such local-global associative frames ensure that GGNNs meet the SO(3) invariance requirement while preserving the crystal's symmetry, enhancing the model's ability to differentiate between distinct atomic structures. Experimental results on multiple datasets demonstrate the effectiveness of SPFrame. We hope that SPFrame provides a new perspective for machine learning and materials science, promoting the specific adaptation of machine learning techniques for materials science applications. Further discussion of SPFrame is provided in Appendix A.10 and Appendix A.11.

## Acknowledgments

This work was partially supported by the Research Grants Counil (RGC) of the Hong Kong (HK) SAR (Grant No. 15208725 and 15208222), the Young Scientists Fund of National Natural Science Foundation of China (NSFC) (Grant No. 62206235), and the Hong Kong Polytechnic University (Grant No. A0046682 and P0057774).

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

# A Appendix / supplemental materials

## A.1 Theoretical Justification of the Superiority of SPFrame over Local Frame

We provide a mutual information-based proof to explain why the SPFrame approach yields more informative node/atom representations than the local frame method. Let a crystal structure be represented as a graph with $N$ atoms, and denote the node/atom feature corresponding to atom $i$ as $\boldsymbol{f}_i$. The complete sets of node or atom features under two different framing schemes are defined as:

$$A = \{\boldsymbol{f}_{local,i} | i = 1, 2, .., N\}, B = \{\boldsymbol{f}_{sp,i} | i = 1, 2, .., N\}, \tag{7}$$

where $A$ represents node/atom features obtained using the local frame method, and $B$ corresponds to those derived using the SPFrame method.

We assume that the atoms with indices $p, q$ ($1 < p, q < N$) are symmetry-equivalent. The use of the local frame results in identical local environments for atoms $p$ and $q$ (as illustrated in Figure 2). Consequently, after message passing via Equation 3, we have

$$\boldsymbol{f}_{local,p} = \boldsymbol{f}_{local,q}, \tag{8}$$

which reduces the number of distinguishable atom representations. Denoting the cardinality as $|\cdot|$, we obtain

$$|A| = N - 1 < N. \tag{9}$$

In contrast, the SPFrame approach preserves the differences in the local environments of atoms $p$ and $q$. After message passing via Equation 5, the node/atom representations satisfy

$$\boldsymbol{f}_{local,p} \neq \boldsymbol{f}_{local,p} \implies |B| = N. \tag{10}$$

For scalar crystal property prediction, the final node features are first aggregated to get a global graph-level representation, which is then passed through a regression head. We denote the prediction target as $Y \in \mathcal{Y}$. The neural network induces the following mapping:

$$h_{local} : A \mapsto \mathcal{Y}, \quad h_{sp} : B \mapsto \mathcal{Y}. \tag{11}$$

Since $|B| > |A|$, there exists a surjective mapping $g_1 : B \mapsto A$, such that $h_{local} \circ g_1 = h_{sp}$. However, an injective mapping $g_2 : A \mapsto B$ does not exist in general due to loss of distinguishability in $|A|$. Consequently, the information flow can be described via the following Markov chain:

$$Y \rightarrow B \rightarrow A. \tag{12}$$

Applying the data processing inequality to this chain yields

$$I(Y; B) \geq I(Y; A), \tag{13}$$

with equality if and only if

$$I(B; Y|A) = 0 \quad \text{and} \quad Y \rightarrow A \rightarrow B, \tag{14}$$

i.e., the chain $Y \rightarrow A \rightarrow B$ is also valid. However, the absence of a mapping $g_2 : A \mapsto B$ implies that this reverse chain cannot be constructed. Therefore, the inequality is strict:

$$I(Y; B) > I(Y; A), \tag{15}$$

This result implies that the node/atom representations obtained via SPFrame retain strictly higher mutual information with the target variable $Y$ than those obtained via the local frame method. In other words, SPFrame-based features preserve more task-relevant information.

## A.2 Proof of SE(3) invariance

This work also ensures SE(3) invariance [15, 16]. Let atoms $i$ and $j$ be two neighboring atoms, with positions denoted by $\boldsymbol{x}_i$ and $\boldsymbol{x}_j$, respectively. In the message passing formulation of Equation 5, the edge scalar feature $e_{ij}$ can be expressed as $||\boldsymbol{x}_i - \boldsymbol{x}_j||$, and the directional vector $\widehat{e}_{ij}$ can be expressed as $\boldsymbol{x}_i - \boldsymbol{x}_j$. After applying a global rotation $\mathbf{Q}$ and translation $\mathbf{t}$, we have

$$\boldsymbol{f}_i^{(k)} = \psi^{(k)}\left(\boldsymbol{f}_i^{(k-1)}, \sum_{j \in \mathcal{N}(i)} \phi^{(k)}\left(\boldsymbol{f}_i^{(k-1)}, \boldsymbol{f}_j^{(k-1)}, \boldsymbol{e}_{ij}, \widehat{\boldsymbol{e}}_{ij}\mathbf{F}_{\text{global}}^{\top}\mathbf{F}_{\text{INV},i}^{\top}\right)\right)$$

$$= \psi^{(k)}\left(\boldsymbol{f}_i^{(k-1)}, \sum_{j \in \mathcal{N}(i)} \phi^{(k)}\left(\boldsymbol{f}_i^{(k-1)}, \boldsymbol{f}_j^{(k-1)}, ||\boldsymbol{x}_i - \boldsymbol{x}_j||, (\boldsymbol{x}_i - \boldsymbol{x}_j)\mathbf{F}_{\text{global}}^{\top}\mathbf{F}_{\text{INV},i}^{\top}\right)\right)$$

Apply $\mathbf{Q}$ and $\mathbf{t}$:

$$= \psi^{(k)}\left(\boldsymbol{f}_i^{(k-1)}, \sum_{j \in \mathcal{N}(i)} \phi^{(k)}\left(\boldsymbol{f}_i^{(k-1)}, \boldsymbol{f}_j^{(k-1)}, ||(\boldsymbol{x}_i\mathbf{Q} - \mathbf{t}) - (\boldsymbol{x}_j\mathbf{Q} - \mathbf{t})||, ((\boldsymbol{x}_i\mathbf{Q} - \mathbf{t}) - (\boldsymbol{x}_j\mathbf{Q} - \mathbf{t}))(\mathbf{F}_{\text{global}}\mathbf{Q})^{\top}\mathbf{F}_{\text{INV},i}^{\top}\right)\right)$$

$$= \psi^{(k)}\left(\boldsymbol{f}_i^{(k-1)}, \sum_{j \in \mathcal{N}(i)} \phi^{(k)}\left(\boldsymbol{f}_i^{(k-1)}, \boldsymbol{f}_j^{(k-1)}, ||\boldsymbol{x}_i - \boldsymbol{x}_j||, (\boldsymbol{x}_i - \boldsymbol{x}_j)\mathbf{Q}\mathbf{Q}^{\top}\mathbf{F}_{\text{global}}^{\top}\mathbf{F}_{\text{INV},i}^{\top}\right)\right)$$

$$= \psi^{(k)}\left(\boldsymbol{f}_i^{(k-1)}, \sum_{j \in \mathcal{N}(i)} \phi^{(k)}\left(\boldsymbol{f}_i^{(k-1)}, \boldsymbol{f}_j^{(k-1)}, \boldsymbol{e}_{ij}, \widehat{\boldsymbol{e}}_{ij}\mathbf{F}_{\text{global}}^{\top}\mathbf{F}_{\text{INV},i}^{\top}\right)\right)$$

(16)

Applying a rotation $\mathbf{Q}$ and translation $\mathbf{t}$ does not change the expression in Equation 5. Therefore, Equation 5 is unaffected by rotation and translation, indicating that it is SE(3)-invariant.

## A.3 Implementation of Global Frame in SPFrame

As outlined in Section 2.1, the entire crystal structure can be represented by its unit cell. When the entire crystal structure undergoes a rotation, the unit cell also changes accordingly. Therefore, the global frame can be computed from the lattice matrix $\mathbf{L} \in \mathbb{R}^{3 \times 3}$ of the unit cell [21, 52]. Below, we introduce three commonly used methods for computing the global frame. Each of these methods can be applied to the global frame construction within the SPFrame.

**QR Decomposition [33].** Given that the lattice matrix $\mathbf{L} \in \mathbb{R}^{3 \times 3}$ is invertible, it can be uniquely decomposed via QR decomposition as $\mathbf{L} = \mathbf{QR}$, where the diagonal elements of $\mathbf{R}$ are constrained to be positive. In this decomposition, $\mathbf{Q} \in \mathbb{R}^{3 \times 3}$ is an orthogonal matrix, while $\mathbf{R} \in \mathbb{R}^{3 \times 3}$ is an upper triangular matrix. By applying QR decomposition to $\mathbf{L}$ under this positivity constraint, we obtain the orthogonal matrix $\mathbf{Q}$, which is naturally equivariant under O(3) transformations. To further restrict this equivariance to the SO(3) group, we flip the sign of the first column vector of $\mathbf{Q}$ if necessary to enforce $\det(\mathbf{Q}) = 1$. The resulting matrix, now SO(3)-equivariant, serves as a choice for the global frame $\mathbf{F}_{\text{global}}$ in our proposed SPFrame.

**Polar Decomposition [21, 18].** As an invertible matrix, the lattice matrix $\mathbf{L} \in \mathbb{R}^{3 \times 3}$ can be uniquely decomposed into $\mathbf{L} = \mathbf{QH}$, where $\mathbf{Q} \in \mathbb{R}^{3 \times 3}$ is an orthogonal matrix, $\mathbf{H} \in \mathbb{R}^{3 \times 3}$ is a Hermitian positive semi-definite matrix. By applying polar decomposition to $\mathbf{L}$, we obtain the orthogonal matrix $\mathbf{Q}$, which is naturally equivariant under O(3) transformations. To ensure that $\mathbf{Q}$ is equivariant only under SO(3) transformations, we adjust the sign of its first column vector if needed to enforce $\det(\mathbf{Q}) = 1$. The resulting matrix, now SO(3)-equivariant, can be reliably utilized as the global frame $\mathbf{F}_{\text{global}}$ in our SPFrame.

**Principal Component Analysis (PCA) [9, 37].** For the lattice matrix $\mathbf{L} \in \mathbb{R}^{3 \times 3}$, we first compute the centroid of the lattice vectors as $\mathbf{t} = \frac{1}{n}\mathbf{L1} \in \mathbb{R}^3$, followed by the construction of the covariance matrix $\Sigma = \left(\mathbf{L} - \mathbf{1}\mathbf{t}^{\top}\right)^{\top}\left(\mathbf{L} - \mathbf{1}\mathbf{t}^{\top}\right)$. We then perform eigendecomposition on $\Sigma$ to obtain its eigenvectors $\mathbf{u}_1, \mathbf{u}_2, \mathbf{u}_3$. Assuming the eigenvalues satisfy the condition $\lambda_1 > \lambda_2 > \lambda_3$, the corresponding eigenvectors can be assembled into a $3 \times 3$ orthogonal matrix $\mathbf{U} = [\mathbf{u}_1, \mathbf{u}_2, \mathbf{u}_3]$, which defines one of eight possible O(3)-equivariant frames. To obtain a unique SO(3)-equivariant global frame, we apply Algorithm 2 to resolve the sign ambiguity [37] in $\mathbf{U}$ and enforce $\det(\mathbf{U}) = 1$. The resulting matrix $\mathbf{U}^*$ serves as the global frame $\mathbf{F}_{\text{global}}$ in SPFrame.

## A.4 Implementation of Invariant Local Frame in SPFrame

**Quaternion Generation via SO(3)-Invariant Message Passing.** In this work, we adopt SO(3)-invariant message passing proposed by Yan et al. [56] to generate quaternions for constructing local frames. This process leverages the atom features $\boldsymbol{f}_i$, neighboring atom features $\boldsymbol{f}_j$, and invariant edge features $\boldsymbol{e}_{ij}$ to perform message passing from neighbor atom $j$ to the central atom $i$. The messages

---

**Algorithm 2** Unique SO(3)-equivariant global frame based on eigenvectors

---

**Require:** The orthogonal eigenvector matrices $\mathbf{U} = [\mathbf{u}_1, \mathbf{u}_2, \mathbf{u}_3]$
**Ensure:** The unique SO(3)-equivariant global frame $\mathbf{U}^*$
 1: **for** $i$=1,2 **do**
 2:     Let $j$ be the smallest index such that $u_j \neq 0$
 3:     **if** $u_j > 0$ **then**
 4:         $\mathbf{u}_i^* \leftarrow \mathbf{u}_i$
 5:     **else**
 6:         $\mathbf{u}_i^* \leftarrow -\mathbf{u}_i$
 7:     **end if**
 8: **end for**
 9: **if** $\det([\mathbf{u}_1^*, \mathbf{u}_2^*, \mathbf{u}_3]) > 0$ **then**
10:     $\mathbf{u}_3^* \leftarrow \mathbf{u}_3$
11: **else**
12:     $\mathbf{u}_3^* \leftarrow -\mathbf{u}_3$
13: **end if**
14: $\mathbf{U}^* = [\mathbf{u}_1^*, \mathbf{u}_2^*, \mathbf{u}_3^*]$

---

are aggregated across all neighbors, and the result is combined with the central atom's features $\boldsymbol{f}i$ to produce the quaternion $\boldsymbol{q}_i$.

Specifically, we first calculate three components: the query vector $\boldsymbol{q}_{ij} = \mathrm{LN}_Q(\boldsymbol{f}_i)$, the key vector $\boldsymbol{k}_{ij} = (\mathrm{LN}_K(\boldsymbol{f}_i)|\mathrm{LN}_K(\boldsymbol{f}_j))$, and the value vector $\boldsymbol{v}_{ij} = (\mathrm{LN}_V(\boldsymbol{f}_i)|\mathrm{LN}_V(\boldsymbol{f}_j)|\mathrm{LN}_E(\boldsymbol{f}_{ij}^e))$, where $\mathrm{LN}_Q(\cdot)$, $\mathrm{LN}_K(\cdot)$, $\mathrm{LN}_V(\cdot)$, $\mathrm{LN}_E(\cdot)$ denote the linear layers, and $|$ denote the concatenation. Then, the message form atom $j$ to atom $i$ is computed as:

$$\boldsymbol{\alpha}_{ij} = \frac{\boldsymbol{q}_{ij} \circ \boldsymbol{\xi}_K(\boldsymbol{k}_{ij})}{\sqrt{d_{\boldsymbol{q}_{ij}}}}, \quad \boldsymbol{msg}_{ij} = \mathrm{sigmoid}(\mathrm{BN}(\boldsymbol{\alpha}_{ij})) \circ \boldsymbol{\xi}_V(\boldsymbol{v}_{ij}), \tag{17}$$

where $\boldsymbol{\xi}_K, \boldsymbol{\xi}_V$ represent mappings applied to the key and value vectors, respectively, and the operators $\circ$ denote the Hadamard product, BN refers to the batch normalization layer, and $\sqrt{d_{\boldsymbol{q}_{ij}}}$ indicates the dimensionality of $\boldsymbol{q}_{ij}$. Finally, the quaternion $\boldsymbol{q}_i$ is generated as:

$$\boldsymbol{msg}_i = \sum_{j \in \mathcal{N}_i} \boldsymbol{msg}_{ij}, \quad \boldsymbol{q}_i = \boldsymbol{\xi}_{msg}(\mathrm{LN}_{msg}(\boldsymbol{f}_i + \mathrm{BN}(\boldsymbol{msg}_i))), \tag{18}$$

where $\boldsymbol{\xi}_{msg}(\cdot)$ denotes the softplus activation function amd $\mathrm{LN}_{msg}(\cdot)$ denote the linear layer.

**Quaternion to Rotation Matrix Conversion.** Quaternions, while originating in pure mathematics, are extensively used for representing and computing 3D rotations [60, 42, 11, 46]. A unit quaternion, represented by four real-valued components, encodes a 3D rotation by specifying a rotation axis and an associated rotation angle [11]. Therefore, we normalize the network's 4-dimensional output to obtain a unit quaternion, which is then converted into a rotation matrix [42].

### A.5  Backbone with SPFrame

The eComFormer has demonstrated strong performance across a wide range of crystal property prediction tasks [57]. This model integrates a node-wise transformer layer and a node-wise equivariant updating layer to capture complex geometric relationships within the crystal structure. Given that the node-wise equivariant updating layer operates on equivariant edge features, we incorporate SPFrame into each of these layers. Furthermore, we append a node-wise equivariant updating layer following each node-wise transformer layer. The detailed network architecture is provided below.

**Node-wise transformer layer in eComFormer.** The node-wise transformer layer in eComFormer updates node invariant features $\boldsymbol{f}_i$ through a message-passing mechanism. This layer integrates three types of information: the node features $\boldsymbol{f}_i$, neighboring node features $\boldsymbol{f}_j$, and invariant edge embeddings $\boldsymbol{f}_{ij}^e$. The update process follows a transformer-style architecture. Fristly, the message from node $j$ to node $i$ is encoded using three projected features query $\boldsymbol{q}_{ij} = \mathrm{LN}_Q(\boldsymbol{f}_i)$, key $\boldsymbol{k}_{ij} = (\mathrm{LN}_K(\boldsymbol{f}_i)|\mathrm{LN}_K(\boldsymbol{f}_j))$, and value feature $\boldsymbol{v}_{ij} = (\mathrm{LN}_V(\boldsymbol{f}_i)|\mathrm{LN}_V(\boldsymbol{f}_j)|\mathrm{LN}_E(\boldsymbol{f}_{ij}^e))$, where $\mathrm{LN}_Q(\cdot)$,

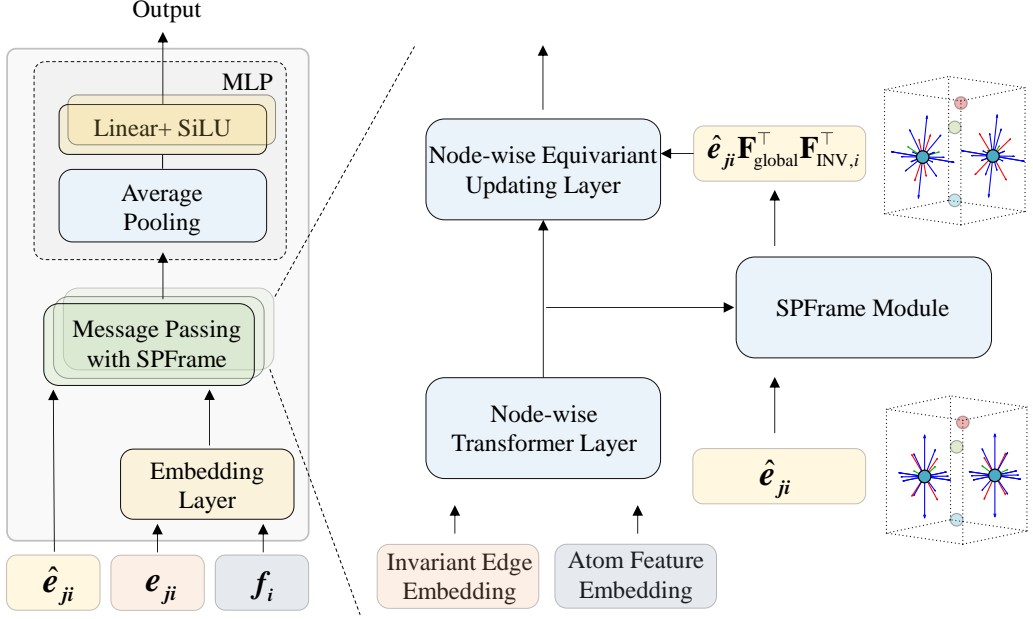

Figure 4: The detailed architectures of eComFormer with SPFrame.

$\text{LN}_K(\cdot), \text{LN}_V(\cdot), \text{LN}_E(\cdot)$ denote the linear transformations, and $|$ denote the concatenation. Then, the attention mechanism computes:

$$\boldsymbol{\alpha}_{ij} = \frac{\boldsymbol{q}_{ij} \circ \boldsymbol{\xi}_K(\boldsymbol{k}_{ij})}{\sqrt{d_{\boldsymbol{q}_{ij}}}}, \boldsymbol{msg}_{ij} = \text{sigmoid}(\text{BN}(\boldsymbol{\alpha}_{ij})) \circ \boldsymbol{\xi}_V(\boldsymbol{v}_{ij}), \tag{19}$$

where $\boldsymbol{\xi}_K, \boldsymbol{\xi}_V$ are nonlinear transformations, and the operators $\circ$ denote the Hadamard product. $\text{BN}(\cdot)$ refers to the batch normalization layer, and $\sqrt{d_{\boldsymbol{q}_{ij}}}$ indicates the dimensionality of $\boldsymbol{q}_{ij}$. Then, node feature $\boldsymbol{f}_i$ is updated as follows,

$$\boldsymbol{msg}_i = \sum_{j \in \mathcal{N}_i} \boldsymbol{msg}_{ij}, \boldsymbol{f}_i^{\text{new}} = \boldsymbol{\xi}_{msg}(\boldsymbol{f}_i + \text{BN}(\boldsymbol{msg}_i)), \tag{20}$$

where $\boldsymbol{\xi}_{msg}(\cdot)$ denoting the softplus activation function.

**Node-wise equivariant updating layer using SPFrame.** The node-wise equivariant updating layer in eComFormer employs two tensor product (TP) layers [12] to effectively capture geometric features. The equivalent edge feature $\boldsymbol{e}_{ji}$ is embedded using spherical harmonics, with the representations given by $\mathbf{Y}_0(\widehat{\boldsymbol{e}}_{ji}) = c_0, \mathbf{Y}_1(\widehat{\boldsymbol{e}}_{ji}) = c_1 * \frac{\widehat{\boldsymbol{e}}_{ji}}{||\widehat{\boldsymbol{e}}_{ji}||_2} \in \mathbb{R}^3$ and $\mathbf{Y}_2(\widehat{\boldsymbol{e}}_{ji}) \in \mathbb{R}^5$. These harmonics form the input features to the TP layers.

Therefore, we apply the SPFrame to the equivariant edge features before embedding them into spherical harmonics. Specifically, the first TP layer is defined as:

$$\boldsymbol{f}_{i,0}^l = \boldsymbol{f}_i^{l'} + \frac{1}{|\mathcal{N}_i|} \sum_{j \in \mathcal{N}_i} \mathbf{TP}_0(\boldsymbol{f}_j^{l'}, \mathbf{Y}_0(\widehat{\boldsymbol{e}}_{ji}\mathbf{F}_{\text{global}}^{\top}\mathbf{F}_{\text{INV},i}^{\top})),$$

$$\boldsymbol{f}_{i,\lambda}^l = \frac{1}{|\mathcal{N}_i|} \sum_{j \in \mathcal{N}_i} \mathbf{TP}_\lambda(\boldsymbol{f}_j^{l'}, \mathbf{Y}_\lambda(\widehat{\boldsymbol{e}}_{ji}\mathbf{F}_{\text{global}}^{\top}\mathbf{F}_{\text{INV},i}^{\top})), \lambda \in \{1, 2\}, \tag{21}$$

where $\boldsymbol{f}_i^{l'}$ is the linearly transformed atom feature derived from $\boldsymbol{f}_i^l$, $|\mathcal{N}_i|$ denotes the number of neighboring atoms of atom $i$, and $\mathbf{TP}_\lambda$ denotes the TP layer corresponding to rotation order $\lambda$.

The second TP layer further aggregates the directional features across multiple orders as follows:

$$\boldsymbol{f}_i^{l*} = \frac{1}{|\mathcal{N}_i|}(\sum_{j \in \mathcal{N}_i} \sum_{\lambda = 0, 1, 2} \mathbf{TP}_0(\boldsymbol{f}_{j,\lambda}^l, \mathbf{Y}_\lambda(\widehat{\boldsymbol{e}}_{ji}\mathbf{F}_{\text{global}}^{\top}\mathbf{F}_{\text{INV},i}^{\top}))) \tag{22}$$

Finally, the outputs from the two TP layers are combined through both linear and nonlinear transformations to produce the updated atom feature $\boldsymbol{f}_{i,updated}^{l}$:

$$\boldsymbol{f}_{i,updated}^{l} = \boldsymbol{\sigma}(\mathrm{BN}(\boldsymbol{f}_i^{l*})) + \mathrm{LN}(\boldsymbol{f}_i^{l}), \tag{23}$$

where $\boldsymbol{\sigma}(\cdot)$ denotes a nonlinear transformation consisting of two softplus layers with an intervening linear layer, while $\mathrm{BN}(\cdot)$ and $\mathrm{LN}(\cdot)$ represent batch normalization and a linear layer, respectively.

**Overall architecture.** The overall architecture is illustrated in Fig. 4. The key components of the network are summarized as follows. The architecture begins with embedding layers for node and edge features, followed by a series of stacked message passing modules. Each module consists of a node-wise transformer layer, a node-wise equivariant update layer, and a SPFrame construction block. The network concludes with a global average pooling layer and a multi-layer perceptron (MLP) for property prediction. Notably, drawing inspiration from recent findings [19], which demonstrate that dynamically constructing frames at intermediate layers significantly enhances both model expressiveness and prediction accuracy, we integrate SPFrame construction modules at multiple stages throughout the network. Furthermore, since eComFormer produces SO(3)-invariant outputs, the global frame $\mathbf{F}_{\mathrm{global}}$ in Equation 5 can be set as the identity matrix. This simplification enables a more efficient implementation of our approach.

## A.6 More Related Works

**Other approaches for improving prediction accuracy.** In crystal property prediction tasks, in addition to frame-based methods (which can also be regarded as a form of representation learning), pretraining [4, 6, 48] and representation learning [17, 5, 35, 39] are two other important approaches for improving prediction accuracy.

Pretraining methods primarily focus on improving the backbone network architecture. CrysXPP [4] designs an autoencoder for self-supervised pretraining, capturing key structural and chemical features from large amounts of unlabeled crystal graph data to reduce prediction errors. CrysGNN [6] introduces a specialized pretrained GNN framework that combines feature reconstruction, connectivity reconstruction, and contrastive learning across different crystal systems. CrysDiff [48] employs a diffusion-based pretraining approach, where the pretraining phase reconstructs crystal structures via a diffusion process to learn the underlying edge distribution, and the fine-tuning phase generates target property values guided by structural data.

In contrast, representation learning focuses on constructing more expressive representations of crystal structures. Beyond commonly used bond angle information for encoding directionality, ALIGNN-d [17] incorporates dihedral angles, achieving a memory-efficient graph representation that captures the full atomic geometry. CrysMMNet [5] integrates textual material descriptions into the crystal graph to encode global structural information, leading to richer and more robust representations. Geom3D [35] systematically benchmarks various geometric encoding strategies, including spherical harmonics, frame-based bases, and angle-based features. CrysAtom [39] learns distributed atomic representations in an unsupervised manner from unlabeled crystal data, significantly improving downstream property prediction.

**Incorporating crystal symmetry into method design.** Crystal symmetry is a fundamental property of crystalline materials. Most existing methods for property prediction have only limited utilization of this symmetry, while many generative approaches explicitly leverage it to improve model performance. The core idea of these generative approaches is to simplify the data to be generated by exploiting crystal symmetry. Below, we introduce several representative methods.

DiffCSP++ [21]: Owing to crystal symmetry, the lattice matrix elements in different space groups are subject to specific constraints. DiffCSP++ generates only the unconstrained lattice elements, simplifying the generation process. It also reconstructs atomic fractional coordinates and element types by deriving symmetry-equivalent atoms from a single representative atom using symmetry operations. This ensures that the generated crystals strictly satisfy space group constraints.

SymmCD [28]: SymmCD generates only the asymmetric unit rather than the full lattice matrix, outputting its unit parameters. Unlike DiffCSP++, which generates complete lattice matrices using predefined templates, SymmCD demonstrates experimentally that DiffCSP++ may limit structural diversity and novelty.

Wyckoff Transformer [24]: Wyckoff Transformer is an autoregressive generative method distinct from diffusion-based approaches [21, 28, 25]. For symmetry-related atoms, it generates only discrete attributes such as space group, element type, site symmetry, and enumeration. The complete crystal structure is reconstructed by combining these discrete attributes with energy relaxation.

WyckoffDiff [25]: Given a space group and Wyckoff positions, WyckoffDiff predicts the probability distribution of atom types occupying each position. This approach resembles UniMat [59], which predicts elemental probabilities from the periodic table, but WyckoffDiff explicitly embeds symmetry constraints into the generative process.

Our method, in contrast, is designed for scalar property prediction rather than generative modeling. It primarily addresses the limitation of local frames, where atoms at symmetry-equivalent positions share identical local environments, leading to indistinguishable node features and information loss. To mitigate this, we design the frame by combining an invariant local frame with an equivariant global frame shared across the atomic system. This design preserves crystal symmetry and ensures that atoms at symmetry-equivalent positions maintain symmetric yet distinguishable local environments, allowing their node features to remain discriminative.

### A.7 Frame Baseline

**SO(3) equivariant frame constructed based on Gram-Schmidt orthogonalization** [50, 34] Inspired by previous work [44, 50], we construct an equivariant frame by predicting two equivariant vectors using the Schmidt orthogonalization method, and use this frame as a baseline for comparison with the proposed Symmetry-preserving frame in this paper. Specifically, the two equivariant vectors $\mathbf{v}_{i,1}, \mathbf{v}_{i,2} \in \mathbb{R}^3$ are predicted as follows:

$$\mathbf{v}_{i,k} = \sum_{j \in \mathcal{N}(i)} \phi_k\left(\boldsymbol{f}_i, \boldsymbol{f}_j, \boldsymbol{e}_{ij}\right) \widehat{\boldsymbol{e}}_{ij}, \quad k \in \{1, 2\}, \tag{24}$$

where $\boldsymbol{f}_i$ denotes the feature vector of atom $i$, $\boldsymbol{e}_{ij}$ represents SO(3)-invariant edge features, and $\widehat{\boldsymbol{e}}_{ij}$ corresponds to SO(3)-equivariant edge features. $\phi_k(\cdot)$ is the message function from Yan et al. [56]. The rotation matrix is then constructed using Gram-Schmidt orthogonalization as follows:

$$\bar{\mathbf{v}}_{i,1} = \frac{\mathbf{v}_{i,1}}{\|\mathbf{v}_{i,1}\|}, \quad \mathbf{v}'_{i,2} = \mathbf{v}_{i,2} - (\mathbf{n}_{i,1} \cdot \mathbf{v}_{i,2})\bar{\mathbf{v}}_{i,1}, \quad \bar{\mathbf{v}}_{i,2} = \frac{\mathbf{v}'_{i,2}}{\|\mathbf{v}'_{i,2}\|}$$
$$\bar{\mathbf{v}}_{i,3} = \bar{\mathbf{v}}_{i,1} \times \bar{\mathbf{v}}_{i,2}, \quad \mathbf{F}_{\text{INV},i} = [\bar{\mathbf{v}}_{i,1}, \bar{\mathbf{v}}_{i,2}, \bar{\mathbf{v}}_{i,3}]^\top \tag{25}$$

**SPFrame constructed based on Gram-Schmidt orthogonalization** During the construction of the SPFrame, we need to establish an invariant local frame and an equivariant global frame. Similar to Eq. 24, we predict two invariant vectors using only invariant edge features:

$$\mathbf{v}_{i,k} = \sum_{j \in \mathcal{N}(i)} \phi_k\left(\boldsymbol{f}_i, \boldsymbol{f}_j, \boldsymbol{e}_{ij}\right), \quad k \in \{1, 2\}. \tag{26}$$

The rotation matrix $\mathbf{F}_{\text{INV},i}$ is then constructed using Eq. 25. The equivariant global frame $\mathbf{F}_{\text{global}}$ is still derived using the method described in Appendix A.3. Ultimately, this yields the Symmetry-preserving frame $\mathbf{F}_{\text{INV},i}\mathbf{F}_{\text{global}}$ based on Gram-Schmidt orthogonalization.

**Incorporating angular information.** When computing the local frame using the Gram-Schmidt orthogonalization method as defined in Equation 25, the intrinsic symmetry of the crystal can lead to cases during training where the vectors $\mathbf{v}_{i,1}$ and $\mathbf{v}_{i,2}$ become collinear. This collinearity prevents the Gram-Schmidt orthogonalization method from producing a valid local frame.

To overcome this limitation, we incorporate angular information into Equation 25. Specifically, for each atom within the unit cell, we first compute the frame (such as PCA frame in Appendix A.3) of the equivariant edge vectors and compute the frame of the vectors in the lattice matrix. These vectors are then transformed into invariant representations. Next, we calculate the angles [57] between the transformed edge vectors and the transformed lattice vectors. These angle-based features are then integrated into the invariant edge features [57], enhancing the robustness of the frame construction.

## A.8   Training Settings

In this subsection, we provide the detailed hyperparameter settings for backbone integrated with SPFrame across different tasks. For the network architecture, the backbone follows the parameter settings outlined in the original paper [57], such as those for the graph construction and the embedding layers. The training hyperparameters are as follows.

**JARVIS: formation energy.** For the eComFormer backbone, the network is trained using L1 loss with the Adam optimizer [26] for 500 epochs, employing the Onecycle scheduler [47] with a pct_start of 0.3 and an initial learning rate of 0.0005. The network consists of 2 message passing layers and 3 SPFrame modules. Each message passing layer is equipped with one SPFrame module, and an additional SPFrame module is placed before the first message passing layer. The intermediate features, such as node features and invariant edge features, are set to 256 dimensions, and the batch size is set to 64. For the iComFormer backbone, the network is trained using L1 loss with the Adam optimizer for 700 epochs, employing the Onecycle scheduler with a pct_start of 0.3 and an initial learning rate of 0.001. The network consists of a total of 4 message passing layers, each equipped with an SPFrame module except for the final layer. The dimensionality of all features is set to 256, and the batch size is 64.

**JARVIS: band gap (OPT).** For the eComFormer backbone, the network is trained using the L1 loss function and the Adam optimizer for 500 epochs. A cosine with warmup scheduler is employed [54], with an initial learning rate of 0.001 and a warmup phase corresponding to 5% of the total training steps. The network consists of a total of 2 message passing layers, each equipped with an SPFrame module. The feature dimension is set to 128, and the batch size is 64. For the iComFormer backbone, the network is trained using the L1 loss function and the Adam optimizer for 500 epochs. A cosine with warmup scheduler is employed, with an initial learning rate of 0.001 and a warmup phase corresponding to 5% of the total training steps. The network consists of a total of 4 message passing layers, each equipped with an SPFrame module except for the final layer. The feature dimension is set to 128, and the batch size is 64.

**JARVIS: band gap (MBJ).** For the eComFormer backbone, the network is trained using the L1 loss function and the Adam optimizer for 500 epochs. A cosine with warmup scheduler is employed, with an initial learning rate of 0.003 and a warmup phase corresponding to 5% of the total training steps. The network consists of a total of 2 message passing layers, each equipped with an SPFrame module. The feature dimension is set to 128, and the batch size is 64. For the iComFormer backbone, the network is trained using L1 loss with the Adam optimizer for 1000 epochs, employing the Onecycle scheduler with a pct_start of 0.3 and an initial learning rate of 0.001. The network consists of a total of 4 message passing layers, each equipped with an SPFrame module except for the final layer. The feature dimension is set to 256, and the batch size is 64.

**JARVIS: total energy.** For the eComFormer backbone, the network is trained using L1 loss with the Adam optimizer for 1000 epochs, employing the Onecycle scheduler with a pct_start of 0.3 and an initial learning rate of 0.001. The network consists of a total of 2 message passing layers, each equipped with an SPFrame module. The feature dimension is set to 128, and the batch size is 32. For the iComFormer backbone, the network is trained using L1 loss with the Adam optimizer for 1000 epochs, employing the Onecycle scheduler with a pct_start of 0.3 and an initial learning rate of 0.001. The network consists of a total of 4 message passing layers, each equipped with an SPFrame module except for the final layer. The feature dimension is set to 256, and the batch size is 64.

**JARVIS: Ehull.** For the eComFormer backbone, the network is trained using L1 loss with the Adam optimizer for 500 epochs, employing the Onecycle scheduler with a pct_start of 0.3 and an initial learning rate of 0.001. The network consists of a total of 2 message passing layers, each equipped with an SPFrame module. The feature dimension is set to 128, and the batch size is 64. For the iComFormer backbone, the network is trained using L1 loss with the Adam optimizer for 1000 epochs, employing the Onecycle scheduler with a pct_start of 0.3 and an initial learning rate of 0.001. The network consists of a total of 4 message passing layers, each equipped with an SPFrame module except for the final layer. The feature dimension is set to 128, and the batch size is 64.

**JARVIS: bulk modulus.** The network is trained using the L1 loss function and the Adam optimizer for 500 epochs. A cosine with warmup scheduler is employed, with an initial learning rate of 0.001 and a warmup phase corresponding to 5% of the total training steps. The network consists of a total

of 2 message passing layers, each equipped with an SPFrame module. The feature dimension is set to 128, and the batch size is 64.

**JARVIS: shear modulus.** The network is trained using the L1 loss function and the Adam optimizer for 500 epochs. A cosine with warmup scheduler is employed, with an initial learning rate of 0.001 and a warmup phase corresponding to 5% of the total training steps. The network consists of a total of 3 message passing layers, each equipped with an SPFrame module. The feature dimension is set to 128, and the batch size is 64.

**MP: formation energy.** The network is trained using L1 loss with the Adam optimizer for 500 epochs, employing the Onecycle scheduler with a pct_start of 0.3 and an initial learning rate of 0.001. The network consists of a total of 2 message passing layers, each equipped with an SPFrame module. The feature dimension is set to 196, and the batch size is 32.

**MP: band gap.** The network is trained using L1 loss with the Adam optimizer for 500 epochs, employing the Onecycle scheduler with a pct_start of 0.3 and an initial learning rate of 0.001. The network consists of a total of 3 message passing layers, each equipped with an SPFrame module. The feature dimension is set to 128, and the batch size is 32.

**MP: bulk moduli.** The network is trained using L1 loss with the Adam optimizer for 500 epochs, employing the Onecycle scheduler with a pct_start of 0.3 and an initial learning rate of 0.001. The network consists of a total of 4 message passing layers, each equipped with an SPFrame module. The feature dimension is set to 512, and the batch size is 64.

**MP: shear moduli.** The network is trained using MSE loss with the Adam optimizer for 500 epochs, employing the Onecycle scheduler with a pct_start of 0.3 and an initial learning rate of 0.001. The network consists of a total of 4 message passing layers, each equipped with an SPFrame module. The feature dimension is set to 128, and the batch size is 64.

### A.9   Additional Experimental Results

To further demonstrate the generality of SPFrame, we conducted additional experiments by integrating SPFrame with iComFormer [57]. Specifically, iComFormer consists of the node-wise transformer layer and the edge-wise transformer layer. In our implementation, an SPFrame construction block was added after each node-wise transformer layer. The resulting frame was applied to the interatomic edge vectors, and then the angles between these edge vectors and the lattice matrix were computed before being fed into the edge-wise transformer layer for edge feature updates. Table 4 presents the experimental results of combining iComFormer with SPFrame on the JARVIS dataset. Because iComFormer serves as a more powerful backbone, the combined model achieves superior performance compared to the model using eComFormer as the backbone across most prediction tasks.

Table 4: Additional property prediction results on the JARVIS dataset.

| Method | Form. energy eV/atom | Total energy eV/atom | Bandgap (OPT) eV | Bandgap (MBJ) eV | E hull eV |
|---|---|---|---|---|---|
| Crystalframer | 0.0263 | 0.0279 | 0.117 | 0.242 | 0.047 |
| eComFormer | 0.0284 | 0.0315 | 0.124 | 0.283 | 0.044 |
| —w/ SO(3)-equivariant Gram-Schmidt local frame | 0.0285 | 0.0296 | 0.115 | 0.271 | 0.043 |
| **—w/ Quaternion-based SPFrame (ours)** | 0.0261 | 0.0276 | 0.107 | **0.239** | **0.042** |
| iComFormer | 0.0272 | 0.0288 | 0.122 | 0.26 | 0.047 |
| —w/ SO(3)-equivariant Gram-Schmidt local frame | 0.0275 | 0.0287 | 0.112 | 0.255 | 0.045 |
| **—w/ Quaternion-based SPFrame (ours)** | **0.0250** | **0.0259** | **0.106** | 0.251 | **0.042** |

Table 5 presents the experimental results for bulk modulus and shear modulus prediction on the JARVIS dataset. CrystalFramer achieves the best performance on bulk modulus, consistent with the results observed on the MP dataset, while our method slightly outperforms CrystalFramer on shear modulus.

### A.10   Limitations

This work investigates the issue that conventional local frame methods, when applied to crystal structures, may inadvertently disrupt the intrinsic symmetry of the crystal. To address this problem, we

Table 5: Additional bulk and shear modulus prediction results on the JARVIS dataset.

| Method | Bulk Modulus (Kv) | Shear Modulus (Gv) |
|---|---|---|
| Matformer | 11.21 | 10.76 |
| CrysGNN [6] | 10.99 | 9.800 |
| CrysDiff [48] | 9.875 | 9.193 |
| Crystalframer | **8.876** | 8.999 |
| eComFormer | 9.777 | 9.435 |
| —w/ SO(3)-equivariant Gram-Schmidt local frame | 9.855 | 9.689 |
| **—w/ Quaternion-based SPFrame (ours)** | 9.357 | **8.963** |

propose SPFrame. Comparative experiments against conventional local frame approaches demonstrate the effectiveness of SPFrame. However, while empirical results validate the benefits of SPFrame, this study does not provide a quantitative theoretical analysis of how symmetry breaking impacts the accuracy of crystal property prediction. This question remains unexplored and is closely related to the broader topic of model interpretability in neural networks [4, 30, 31, 53, 29]. Future research could pursue a theoretical framework to quantify the effects of symmetry disruption and further elucidate its influence on prediction performance.

## A.11 Broader Impacts

As a frame-based method tailored for crystal structures, SPFrame enhances the accuracy of prediction models and facilitates the discovery of new materials with desirable properties. Therefore, this work has the potential to make a meaningful impact in the field of materials science. Furthermore, SPFrame offers a new perspective at the intersection of machine learning and materials science. By adapting machine learning techniques to account for the unique characteristics of crystal systems, SPFrame demonstrates how domain-specific modifications can significantly improve model performance. This highlights the importance of developing specialized methodologies to promote the effective application of machine learning in materials science.

