# OpenReview forum: "Local-Global Associative Frames for Symmetry-Preserving Crystal Structure Modeling"
_NeurIPS.cc/2025/Conference — NeurIPS 2025 poster_

### Official Review · Reviewer_nQBH · 2025-06-26

**Clarity:** 3
**Significance:** 3
**Originality:** 3
**Rating:** 5
**Confidence:** 4

**Summary:**

The paper highlights and aims to address the issues with existing global and local frames for modelling crystal structures by proposing a new symmetry-preserving frame (SPFrame) which combines local invariant and global equivariant frames. Next, the authors show that the results improve for the materials property prediction task when SPFrame is used with a state-of-the-art architecture.

**Questions:**

Please refer to the Strengths and Weaknesses section.

**Ethical Concerns:**

["NO or VERY MINOR ethics concerns only"]

**Final Justification:**

I recommend acceptance of this paper for its contribution to improving crystal modelling by enhancing frame creation to handle crystal symmetry.

**Limitations:**

Please refer to the Strengths and Weaknesses section.

**Paper Formatting Concerns:**

Not applicable.

**Quality:**

3

**Strengths And Weaknesses:**

**Strength**:
- The paper highlights and addresses an important problem in crystal modelling. The paper is structured and written.
- The problem with the current methods is discussed in detail, and suitable examples are provided to support the claims, particularly in the case of existing local frames.
- The proposed idea is straightforward to integrate with existing works, making it easy to implement the modifications.
- The authors have benchmarked their method on two datasets, JARVIS and MP, showing that their method is either better or slightly worse than the state-of-the-art. Overall, when a baseline model is compared with the baseline $+$ SPFrame, there are notable improvements.

**Weaknesses**:
- I don't fully understand Figure 3. It seems that the behaviour exhibited in both SPFrame and equivariant local frame is similar. Can you point out the exact differences with the help of the colours present in the figure?
- Can you discuss why the quaternion-derived SPFrame provide a better representation than the Gram-Schmidt-based SPFrame? Are there other ways through which $F_{INV, i}$ could be designed, and if so, are there ablations on them?
- Can the proposed SPFrame be used for other tasks related to crystals, such as crystal structure prediction or materials generation?
- Discussion on related works (DiffCSP++ [1], SymmCD [2], WyckoffTransformer [3], WyckoffDiff [4], etc.), which use the crystal symmetry in the architecture design, is missing. Can the authors compare and contrast these ideas with their proposed SPFrame?
- Can you show improvements with SPFrame for another SOTA model? Currently, it is only demonstrated with eComFormer.

References:
1. Space Group Constrained Crystal Generation. Jiao et al., ICLR 2024.
2. SymmCD: Symmetry-Preserving Crystal Generation with Diffusion Models. Levy et al., ICLR 2025.
3. Wyckoff Transformer: Generation of Symmetric Crystals. Kazeev et al., arXiv 2025.
4. WyckoffDiff-A Generative Diffusion Model for Crystal Symmetry. Kelvinius et al., ICML 2025.

---

> ### Author Rebuttal · Authors · 2025-07-31
>
> We sincerely thank you for your constructive suggestions. Below we will address your questions in detail.
>
> ****
>
> **$\bullet$ Response to weakness 1**:
>
> Thank you for your comments.
>
> In the figure, edges of different colors represent connections between atoms of different types (see the lower left corner of Figure 3). Please note the red and green edges in Figure 3, which are highlighted with glowing effects for better visual distinction. It can be observed that applying the local frame leads to identical local structures (the red and green edges of two different atoms point in the same direction). In contrast, applying SPFrame preserves the differences in local structures (the red and green edges of the two atoms point in different directions). This structural identity in the local frame causes the node/atom features to become identical after message passing, resulting in information loss. For more details, please refer to our response to Reviewer 5W3r's weaknesses.
>
> We hope our response addresses the reviewer’s question.
>
> ****
>
> **$\bullet$ Response to weakness 2**:
>
> We sincerely thank you for your insightful comments. Due to the highly symmetric nature of crystal structures, using the Gram-Schmidt orthogonalization-based rotation matrix construction method (detailed in Appendix A.4) may result in two invariant vectors being nearly identical, leading to redundancy and duplication of information. This can increase the difficulty of network training and hinder performance improvements. In contrast, quaternion-based methods do not suffer from this limitation.
>
> In addition to Gram-Schmidt and quaternion, Ref. [1] proposes a rotation matrix construction method based on singular value decomposition (SVD).
> We conducted supplementary experiments using this method to construct $F_{\text{inv}}$ on the JARVIS dataset, and the results are shown below.
>
> ### Table: Property prediction results on the JARVIS dataset
>
> | Method   | Form. energy (eV/atom) | Total energy (eV/atom) | Bandgap (OPT) (eV) |Bandgap (MBJ) (eV) | E hull (eV) |
> |-----|-----|------|-----|-----|----|
> | eComFormer | 0.0284  | 0.0315 | 0.124|0.283 |0.044|
> | — w/ Gram-Schmidt-based SPFrame (ours)| 0.0268  | 0.0281   | 0.109 | 0.259 |  0.043 |
> | — w/ Quaternion-based SPFrame (ours)  | **0.0261**  | **0.0276** | **0.107** |  **0.239** |**0.042** |
> | — w/ SVD-based SPFrame (ours) | 0.0266 | 0.0281| **0.107**  | 0.258 | **0.042** |
>
> Compared to the quaternion-based approach, the SVD-based method did not yield significant improvements. In [1], the SVD-based method demonstrated better performance on scalar regression tasks (e.g., Experiment 4, where an MLP is used to predict scalar values). However, since our task is more complex—mapping from graph structures to scalar properties, this may account for the performance difference between the quaternion-based and SVD-based methods. A more detailed investigation will be left for future work.
>
>
> We hope our response addresses the reviewer’s question.
>
> [1] Learning with 3D rotations, a hitchhiker’s guide to SO(3). *ICML2024*.
>
> ****
>
> **$\bullet$ Response to weakness 3**:
>
> We sincerely thank you for your constructive comments.
>
> Previously, DiffCSP++ [1] applied the frame method to crystal structure prediction and materials generation, where frames were used to construct invariant representations of crystals, thereby removing the requirement for equivariance in the subsequent denoising model. Our method can serve a similar purpose. Since our approach is learnable frames, a more recent concurrent work - RADM [2] (for molecular generation) offers an insightful application strategy. RADM employs frames to eliminate the need for equivariance in the denoising model, allowing the use of a more advanced non-equivariant diffusion transformer (DiT) as the denoising model to enhance generation quality. Additionally, it leverages an autoencoder to learn richer feature representations in the latent space.
>
> Similarly, our SPFrame can be applied in crystal generation. During autoencoder training, SPFrame can preserve crystal symmetries and enable the learning of more expressive latent features. Moreover, by removing the equivariance constraint from the denoising model, SPFrame allows the adoption of powerful non-equivariant DiT to further improve generation performance.
>
> We hope our response addresses the reviewer’s question.
>
>
> [1] Space Group Constrained Crystal Generation. *ICLR2024*.
>
> [2] Scalable Non-Equivariant 3D Molecule Generation via Rotational Alignment. *ICML2025*.
>
> ****
>
> **$\bullet$ Response to weakness 4**:
>
> We sincerely thank you for your constructive comments.
>
> First, we discuss how these generation methods leverage crystal symmetry during the generation process. The core idea of these approaches is to simplify the data to be generated by exploiting the crystal symmetry (For example, by generating only the asymmetric unit and then reconstructing the full crystal structure using symmetry operations).
>
> DiffCSP++ [1]: Due to crystal symmetry, the elements of the lattice matrix in crystals belonging to different space groups are subject to specific constraints, as described in Table 1 of [1]. Therefore, when generating the lattice matrix, DiffCSP++ only needs to generate the elements that are not restricted by the crystal symmetry, thereby simplifying the generation process.
> For the generation of atomic fractional coordinates and types, crystal symmetry allows further simplification. Since atoms at symmetrically equivalent positions can be derived from a single representative atom, it is sufficient to generate only one such atom, and the rest can be obtained through symmetry operations.
> DiffCSP++ separately and simultaneously generates the lattice, fractional coordinates, and atomic composition under the reduced form of the space group constraint. The generated crystals strictly satisfy the space group constraint.
>
> SymmCD [2]: SymmCD only generates the asymmetric unit and, during the generation process, it produces only the unit parameters of the asymmetric unit rather than the full lattice matrix (as illustrated in Figure 1 of [2]). This approach differs from that of DiffCSP++, which focuses on generating the full lattice matrix and uses predefined structural templates to enforce space group constraints. SymmCD also provides experimental evidence suggesting that this approach in DiffCSP++ (using predefined structural templates) may limit the diversity and novelty of the generated samples.
>
> Wyckoff Transformer [3]: Wyckoff Transformer is an autoregressive generation method, distinct from diffusion-based methods [1,2,4]. For atoms located at symmetric positions, it generates only discrete data such as the space group, atomic element, site symmetry, and enumeration, rather than the explicit atomic coordinates or the crystal lattice matrix. The final atomic structure is obtained by combining generated discrete information with an energy relaxation method.
>
>
> WyckoffDiff [4]: WyckoffDiff, given a space group and Wyckoff positions, directly predicts the probability of the atom type occupying the corresponding position, as shown in Figure 1 of [4]. This approach bears some similarity to [5], where the probability of an element’s occurrence is predicted given the periodic table. However, WyckoffDiff incorporates symmetry constraints into the structure generation process.
>
> Our method is applied to scalar property prediction of crystals, which differs from the use cases of the aforementioned generative methods. Our approach primarily addresses the limitation of local frames, which cause atoms at symmetry-equivalent positions to have identical local structures. This structural uniformity leads to identical node features and consequently results in information loss (see our response to Reviewer 5W3r's Weaknesses). Therefore, when designing the frame, we adopt an associative approach that combines an invariant local frame with an equivariant global frame shared across the atomic system. This design prevents the destruction of crystal symmetry and ensures that atoms at symmetry-equivalent positions retain their symmetric local environments, thereby enabling their node features to remain distinguishable.
>
> [1] Space Group Constrained Crystal Generation. *ICLR2024*.
>
> [2] SymmCD: Symmetry-Preserving Crystal Generation with Diffusion Models. *ICLR2025*.
>
> [3] Wyckoff Transformer: Generation of Symmetric Crystals. *ICML2025*.
>
> [4] WyckoffDiff -- A Generative Diffusion Model for Crystal Symmetry. *ICML2025*.
>
> [5] Scalable Diffusion for Materials Generation. *ICLR2024*.
>
> We will include a discussion above of these works in our paper.
> We hope our response addresses the reviewer’s question.
>
> ****
>
> **$\bullet$ Response to weakness 5**:
>
>
> We sincerely thank you for your valuable comments.
>
> We conducted experiments using iComFormer [1] as the backbone, which outperforms eComFormer. Below, we provided the evaluation of JARVIS dataset. Notably, our solution outperforms or at least is on par with the baselines (Crystalframer). The preliminary results are as follows.
>
> ### Table: Property prediction results on the JARVIS dataset
> | Method                                               | Form. energy (eV/atom) | Total energy (eV/atom) | Bandgap (OPT) (eV) | Bandgap (MBJ) (eV) | E hull (eV) |
> |-----|------|-----|-----|-------|------|
> | Crystalframer| 0.0263    | 0.0279| 0.117| 0.242 | 0.047|
> | eComFormer | 0.0284  | 0.0315  | 0.124 | 0.283 | 0.044|
> | — w/ SO(3)-equivariant Gram-Schmidt local frame  | 0.0285 | 0.0296| 0.115| 0.271 | 0.043|
> | **— w/ Quaternion-based SPFrame (ours)**  | 0.0261  | 0.0276 | 0.107  | **0.239** | **0.042**  |
> | iComFormer  | 0.0272 | 0.0288  | 0.122 | 0.261  | 0.047  |
> | — w/ SO(3)-equivariant Gram-Schmidt local frame| runing |runing   | 0.112  |  0.255   | 0.045  |
> | **— w/ Quaternion-based SPFrame (ours)**   | **0.0250**  | **0.0259** | **0.106** | 0.251  | **0.042**  |
>
> We hope our response addresses your concerns.

---

> ### Comment · Reviewer_nQBH · 2025-08-02
> **Response to Authors**
>
> Thank you for your detailed answers and for running additional experiments to show improvements with iComFormer and the benefits of Quaternion-based SPFrame. Also, it would be good to add the explanation corresponding to Figure 3 in the paper. I retain my score and recommend acceptance.

---

> > ### Author Response · Authors · 2025-08-03
> >
> > Dear Reviewer nQBH,
> >
> > We sincerely thank you for your valuable comments and for recommending acceptance. We will add the explanation corresponding to Figure 3 in the paper.
> >
> > Best regards,
> >
> > The Authors

---

### Official Review · Reviewer_7QqS · 2025-06-28

**Clarity:** 3
**Significance:** 3
**Originality:** 2
**Rating:** 4
**Confidence:** 5

**Summary:**

In this paper, the authors addressed an important task in material design: crystal property prediction. A fundamental requirement for crystal property prediction is that the model's output should remain invariant to arbitrary rotations of the input structure. This is essential because crystal structures are inherently equivariant under SO(3) group transformations (i.e., rotations).
The global frame approach can be integrated with any GGNN without imposing constraints on the network architecture, while still ensuring adherence to the SO(3) invariance requirement. However, using only local or only global frames comes with certain limitations.
The authors proposed Symmetry-Preserving Frames for modeling crystal structures, where a combination of local and global associative frames is integrated with GNN for crystal property prediction. This approach helps preserve the inherent symmetry of the crystal.

**Questions:**

Check the weaknesses

**Ethical Concerns:**

["NO or VERY MINOR ethics concerns only"]

**Final Justification:**

NA

**Limitations:**

Check the weaknesses

**Quality:**

3

**Strengths And Weaknesses:**

**Strengths**
- The paper is well-written, with a clearly motivated problem and sufficient background information provided to support the study.
- The authors introduced both global and local frames and integrated them with GNN models. The local frames correspond to invariant local frames for each atom ii, while the global frame is an equivariant frame shared across the entire atomic system. The global frame ensures that the output of the GGNNs remains invariant, whereas the invariant local frames are constructed using a method based on local structural information, ensuring rotational invariance at the atomic level.
- The authors also ensure that atoms pp and qq, which occupy the same type of Wyckoff position, are assigned identical local frames. This guarantees that their transformed representations remain distinguishable while still preserving their relative structural differences.
- Source code is provided

**Weaknesses**

- **Limited Novelty** – While frame-based methodologies have shown strong potential in enforcing equivariance and invariance in geometric deep learning, the authors have primarily applied these existing concepts to the domain of crystal structure prediction. This adaptation, though technically sound, positions the work as more incremental than groundbreaking, as it builds upon well-established techniques rather than introducing fundamentally new methods or insights.

- **Limited Performance Improvement** – When compared to CrystalFramer, the performance gains achieved by the proposed methodology are marginal across both datasets (Table-1/2). This raises concerns about the practical significance and effectiveness of the approach, especially given the added complexity introduced by integrating symmetry-preserving frames.

- Table 2 shows weaker bulk and shear modulus results on the MP dataset. How does the model perform on these properties in the JARVIS dataset?

- **Typo in Equation 1** – There appears to be a typographical error in Equation 1

- Some recent and relevant works in this domain appear to be missing from the related literature discussion. These include:

https://www.nature.com/articles/s41524-022-00716-8

https://www.nature.com/articles/s41524-022-00841-4

https://arxiv.org/abs/2301.05852

https://openreview.net/pdf?id=06jLJiUAKX

https://arxiv.org/pdf/2306.09375

https://ojs.aaai.org/index.php/AAAI/article/view/28748

https://arxiv.org/abs/2409.04737

---

> ### Author Rebuttal · Authors · 2025-07-31
>
> We sincerely thank you for your useful comments. Below we will address your questions in detail.
>
>
> ****
>
> **$\bullet$ Response to weakness 1**:
>
>
> We sincerely thank you for your comments.
>
> Frame-based methodologies are indeed a general geometric learning approach. Our contribution first lies in identifying and articulating the challenges that arise when directly applying general frame methods to crystalline materials (highly symmetric atomic systems). We then propose a targeted solution to address these challenges, which, to the best of our knowledge, has not been considered in prior work.
>
> Moreover, we provide a theoretical justification demonstrating the superiority of SPFrame over conventional local frame methods (please refer to our response to Reviewer 5W3r’s weaknesses), which further strengthens the theoretical significance of our work.
>
> Finally, SPFrame provides a new insight for both machine learning and materials science, namely, that general-purpose machine learning techniques must undergo specific adaptation to effectively address the unique characteristics of materials science problems.
>
>
> We hope our response addresses your concerns.
>
>
>
> ****
>
> **$\bullet$ Response to weakness 2**:
>
> We sincerely thank you for your valuable comments. Our method primarily focuses on enhancing the performance of the backbone (As demonstrated by the performance gains achieved when applying SPFrame to eComFormer). Using a more powerful backbone is theoretically expected to yield even better results. To address the reviewer’s concerns, we conducted additional experiments by replacing the backbone to achieve more significant improvements over the baseline. Specifically, we used iComFormer [1], which outperforms eComFormer, as the new backbone. Below, we provided the evaluation of JARVIS dataset. Notably, our solution outperforms or at least is on par with the baselines (Crystalframer). The preliminary results are as follows.
>
>
>
>
> ### Table: Property prediction results on the JARVIS dataset
>
> | Method                                               | Form. energy (eV/atom) | Total energy (eV/atom) | Bandgap (OPT) (eV) | Bandgap (MBJ) (eV) | E hull (eV) |
> |---------|--------|----------|-----|--------|--------------|
> | Crystalframer      | 0.0263    | 0.0279     | 0.117    | 0.242     | 0.047        |
> | eComFormer       | 0.0284      | 0.0315   | 0.124   | 0.283       | 0.044        |
> | — w/ SO(3)-equivariant Gram-Schmidt local frame      | 0.0285     | 0.0296     | 0.115| 0.271 | 0.043|
> | **— w/ Quaternion-based SPFrame (ours)**  | 0.0261  | 0.0276 | 0.107  | **0.239**           | **0.042**    |
> | iComFormer          | 0.0272     | 0.0288   | 0.122   | 0.261  | 0.047  |
> | — w/ SO(3)-equivariant Gram-Schmidt local frame      | runing    |runing       | 0.112     |  0.255     | 0.045        |
> | **— w/ Quaternion-based SPFrame (ours)**        | **0.0250**  | **0.0259** | **0.106** | 0.251               | **0.042**    |
>
>
>
> We hope our response addresses your concerns.
>
>
> [1] Complete and efficient graph transformers for crystal material property prediction. *ICLR2024*.
>
>
>
>
> ****
>
> **$\bullet$ Response to weakness 3**:
>
>
> We sincerely thank you for your valuable comments. We completed preliminary experiments on bulk and shear modulus properties in the JARVIS dataset, as shown below.
>
>
>
>
> ### Table: Bulk and shear modulus in the JARVIS dataset prediction results
>
>
> | Method          | Bulk Modulus (Kv) |Shear Modulus (Gv) |
> |---------|--------|-------|
> | Matformer    | 11.21  | 10.76|
> | CrysGNN [1]     | 10.99  | 9.800 |
> | CrysDiff [2]     | 9.875  | 9.193|
> | Crystalframer     | **8.876** | 8.999  |
> | eComFormer      |9.777   | 9.435 |
> | — w/ SO(3)-equivariant Gram-Schmidt local frame      | 9.855  |  9.689|
> | **— w/ Quaternion-based SPFrame (ours)**  | 9.357 | **8.963** |
>
> It can be observed from the table that SPFrame significantly improves the performance of the backbone method eComFormer, achieving the best prediction result for Shear Modulus (Gv).
>
>
> [1] CrysGNN: Distilling pre-trained knowledge to enhance property prediction for crystalline materials. *AAAI2023*.
>
> [2] A Diffusion-Based Pre-training Framework for Crystal Property Prediction. *AAAI2024*.
>
>
>
>
>
> ****
>
> **$\bullet$ Response to weakness 4**:
>
>
> Thank you for your careful review. We will revise these typos. Equation 1 was mistakenly written as an expression describing equivariance; we will correct it to properly describe invariance.
>
>
>
>
>
> ****
>
> **$\bullet$ Response to weakness 5**:
>
>
>
> We sincerely appreciate your useful comments. We will include a discussion below of these related works in our paper.
>
> In crystal property prediction tasks, in addition to frame-based methods (which can also be regarded as a form of representation learning), pretraining [1,3,6] and representation learning [2,4,5,7] are two other important approaches for improving prediction accuracy.
>
> Pretraining methods mainly focus on pretraining the backbone network architecture. CrysXPP [1] designs an autoencoder, CrysAE, to perform self-supervised pretraining, thereby capturing essential structural and chemical features from large amounts of unlabeled crystal graph data and reducing prediction errors. CrysGNN is a dedicated pretrained GNN framework that, in addition to common self-supervised strategies such as feature reconstruction used in CrysXPP, incorporates connectivity reconstruction and contrastive learning over different crystal systems, which further enhances prediction performance. CrysDiff [6], on the other hand, is a diffusion-based pretraining method. During the pretraining phase, CrysDiff performs a crystal structure reconstruction task based on a diffusion model to learn the underlying edge distribution of crystal structures. In the fine-tuning phase, the model generates the target property values under the guidance of the structural data.
>
> Unlike pretraining, representation learning primarily focuses on constructing more effective representations of crystal structures. Beyond the commonly used bond angle information to encode directional features, ALIGNN-d [2] incorporates dihedral angles into the crystal representation. This simple extension leads to a memory-efficient graph representation that captures the complete geometry of atomic structures.
> CrysMMNet [4] leverages textual descriptions of materials to encode global structural information into the graph structure, enabling the learning of a more robust and enriched representation of crystalline materials. Geom3D [5] is a benchmarking study on geometric representations, systematically evaluating the effectiveness of various geometric encoding strategies such as spherical harmonics basis, frame-based basis, and angle-based features.
> CrysAtom [7] uses unlabeled crystal data in an unsupervised manner to learn a distributed representation of atoms, which significantly improves the performance of various property prediction models.
>
>
> We hope our response addresses the reviewer’s concerns.
>
>
> [1] CrysXPP: An explainable property predictor for crystalline materials. *npj Computational Materials, 2022*.
>
> [2] Efficient and interpretable graph network representation for angle-dependent properties applied to optical spectroscopy. *npj Computational Materials, 2022*.
>
> [3] CrysGNN: Distilling pre-trained knowledge to enhance property prediction for crystalline materials. *AAAI2023*.
>
> [4] CrysMMNet: Multimodal Representation for Crystal Property Prediction. *UAI2023*.
>
> [5] Symmetry-informed geometric representation for molecules, proteins, and crystalline materials. *NeurIPS2023*.
>
> [6] A Diffusion-Based Pre-training Framework for Crystal Property Prediction. *AAAI2024*.
>
> [7] CrysAtom: Distributed Representation of Atoms for Crystal Property Prediction. *LoG2024*.

---

> > ### Comment · Reviewer_7QqS · 2025-08-02
> >
> > Thank you for addressing all my comments. The results are very convincing and should be integrated into the paper. I have no further questions. I have raised my score.

---

> > > ### Author Response · Authors · 2025-08-03
> > >
> > > Dear Reviewer 7QqS,
> > >
> > > We sincerely thank you for your thoughtful comments and for raising the score. We will incorporate these results into the paper as suggested.
> > >
> > > Best regards,
> > >
> > > The Authors

---

### Official Review · Reviewer_drCP · 2025-07-02

**Clarity:** 4
**Significance:** 2
**Originality:** 3
**Rating:** 5
**Confidence:** 4

**Summary:**

This paper introduces SPFrame, a method for enforcing SO(3) invariance by combining local and global frames. It addresses a limitation of prior equivariant local frame methods, which fail to distinguish atoms in the same Wyckoff position that differ only by rotation. SPFrame resolves this by first aligning SO(3)-equivariant edge features to a global frame to ensure invariance, and then further refining the representation using a local invariant frame to distinguish atoms within the same Wyckoff position. The approach achieves state-of-the-art performance on the MP and JARVIS datasets.

**Questions:**

1. How is translational invariance handled in the proposed method?
2. Have the authors considered using other architectures (besides eComFormer) or providing additional evidence to support the method’s general effectiveness?
3. Although the method achieves SOTA performance, the results are very close to Crystalframer. Have the authors considered reporting metrics over multiple runs to assess statistical significance?
4. (Minor) What exactly is $\hat{e}_{ij}$ in this work? The paper describes it as SO(3)-equivariant edge features, but this is vague. Is it simply the edge vector between atoms i and j?

**Ethical Concerns:**

["NO or VERY MINOR ethics concerns only"]

**Final Justification:**

The authors have clarified their motivation and addressed my concerns about translational invariance. They have also shown that their method works well on both iComFormer and eComFormer. I support the acceptance of this paper.

**Limitations:**

Yes

**Quality:**

3

**Strengths And Weaknesses:**

# Strengths
- The paper is clearly written and easy to follow, with self-contained preliminaries, well-designed figures, and helpful visualizations.
- The proposed method is simple yet appears effective.

# Weaknesses
- The motivation is not entirely convincing. Atoms at the same Wyckoff position share identical local environments, so it seems reasonable for them to have the same representation. The argument that they should be distinguished based on their rotational differences is not convincing.
- The paper lacks a discussion of translational invariance. While the proposed method enforces SO(3) invariance, prior works (e.g., Crystal Framer, CrystalFormer) guarantee full SE(3) invariance.
- Although the method achieves state-of-the-art performance overall, the improvements over the baselines appear marginal (e.g., 0.0261 vs. 0.0263, 0.0172 vs. 0.0171), and it is unclear whether these differences are statistically significant.
- Since the method is only tested with eComFormer, its general effectiveness remains uncertain.

## Typos (and/or minor comments)
- Equation 1: Should be $f_{\theta}(A, QX, QL) = f_{\theta}(A,X,L)$. The current equation is SO(3)-equivariance.
- Line 98: The rightarrow notation seems a bit confusing
- Line 217: CrystalFormer [32] -> Crystalformer
- Line 219: Crystalformer [36] -> CrystalFormer

---

> ### Author Rebuttal · Authors · 2025-07-31
>
> We sincerely thank you for your thoughtful feedback. Below we will address your questions in detail.
>
> ****
>
> **$\bullet$ Response to weakness 1**:
>
> We sincerely thank you for your valuable comments.
> First, it is important to clarify that atoms at the same Wyckoff position share similar local environments, but not identical ones. As illustrated in Figure 2 of our paper (page 4), they can be transformed into the same local environment through symmetry operations such as rotations. Below, we discuss our motivation both from an intuitive and theoretical perspective.
>
> **From an intuitive perspective:** As mentioned in the Introduction of our paper, directional information between atoms is a crucial geometric feature. Leveraging this information can enhance network performance. For example, ComFormer [2] improves upon Matformer [1] by incorporating directional information into the message passing process. However, as shown in Figure 2 of our paper, applying a local frame to ComFormer causes the directional information of symmetry-equivalent atoms p and q to be lost. This results in the node features of atoms p and q becoming identical across all subsequent message passing layers of the network, effectively degrading ComFormer back to Matformer. In contrast, the original ComFormer preserves the directional distinction between p and q, ensuring that their node features remain different in all subsequent layers.
> SPFrame is designed to overcome this limitation of the local frame. It ensures that the node features of symmetry-equivalent atoms p and q remain distinct throughout all layers of message passing, just as in the original ComFormer, thereby preserving essential geometric information.
>
>
> **From a theoretical perspective:** Let a crystal structure be represented as a graph with $N$ atoms. Denote the node/atom feature corresponding to atom $i$ as $f_i$.  We define the complete set of node/atom features under two different framing schemes as follows:
>
> $$A=\\{f_{local,i}|i=1,2,..,N\\}, B=\\{f_{sp,i}|i=1,2,..,N\\},$$
>
> where $A$ corresponds to node/atom features in the network using local frames, and $B$ corresponds to those in the network using the SPFrame approach.
>
> We assume that the atoms with indices $p,q$ ($1<p,q<N$) are symmetry-equivalent.
> The use of the local frame results in identical local environments for atoms p and q (According to the illustration in Figure 2 in our paper, page 4). As a consequence, after message passing via Equation (3) (in our paper, page 4), we can have
>
> $$f_{local,p}=f_{local,q}. $$
>
> Consequently, the effective number of distinguishable node/atom representations is reduced. Denoting the cardinality as $|\cdot|$, we can have:
>
> $$|A|=N-1<N.$$
>
> In contrast, the SPFrame approach preserves the differences in the local environments of atoms p and q. After message passing via Equation (5) (in our paper, page 5), the node/atom representations satisfy:
>
> $$f_{local,p} \neq f_{local,p}  \implies |B|=N.$$
>
>
> For scalar crystal property prediction, the final node features are first aggregated to get global graph-level representation. This graph-level representation is then passed through a regression head. We denote the prediction target as a variable  $Y \in \mathcal{Y}$. The neural network induces the following mapping:
>
>
> $$h_{local}:A \mapsto \mathcal{Y}, \quad h_{sp}:B \mapsto \mathcal{Y}.$$
>
> Since $|B|>|A|$, there exists a surjective mapping $g_1:B \mapsto A$, such that $h_{local}\circ g_1=h_{sp}$. However, an injective mapping $g_2:A \mapsto B$ does not exist in general due to loss of distinguishability in $|A|$. Consequently, the information flow can be described via the following Markov chain:
> $$Y \to B \to A.$$
>
> Applying the data processing inequality to this chain yields:
> $$I(Y;B) \ge I(Y;A),  $$
>
> with equality if and only if:
>
> $$I(B;Y | A) =0 \quad\text{and}\quad Y \to A \to B$$
>
> i.e., the chain $Y \to A \to B$ is also valid. However, the absence of a mapping $g_2:A \mapsto B$ implies that this reverse chain cannot be constructed. Therefore, the inequality is strict:
> $$I(Y;B) \gt I(Y;A),  $$
>
>
> This result implies that the node/atom representations obtained via SPFrame retain strictly higher mutual information with the target variable $Y$ than those obtained via the local frame method. In other words, SPFrame-based features preserve more task-relevant information.
>
>
>
> [1] Periodic graph transformers for crystal material property prediction. *NeurIPS2022*.
>
> [2] Complete and efficient graph transformers for crystal material property prediction. *ICLR2024*.
>
>
>
>
>
>
> ****
>
> **$\bullet$ Response to weakness 2 and question1**:
>
> We sincerely thank you for your constructive comments.
>
> This work also ensures SE(3) invariance. We provide the proof below. Let atoms i and j be two neighboring atoms, with positions denoted by $p_i$ and $p_j$, respectively.
> In the message passing formulation of Equation (5) (in our paper, page 5), the edge scalar feature $e_{ij}$ can be expressed as $||p_i-p_j||$, and the directional vector $\widehat{ e}_{ij}$ can be expressed as $p_i-p_j$. After applying a global rotation $R$ and translation $t$, we have
>
> \begin{align}
> \boldsymbol{f}_i^{(k)} &= \psi^{(k)} ( \boldsymbol{f}_i^{(k-1)}, \sum\_{j \in \mathcal{N}(i)} \phi^{(k)} (\boldsymbol{f}_i^{(k-1)}, \boldsymbol{f}_j^{(k-1)},e\_{ij}, \widehat{e}\_{ij} \mathbf{F}^{\top}\_{\text{global}} \mathbf{F}^{\top}\_{\text{INV},i})) \\\\
> &=  \psi^{(k)} ( \boldsymbol{f}_i^{(k-1)}, \sum\_{j \in \mathcal{N}(i)} \phi^{(k)} (\boldsymbol{f}_i^{(k-1)}, \boldsymbol{f}_j^{(k-1)},||p_i-p_j||, (p_i-p_j) \mathbf{F}^{\top}\_{\text{global}} \mathbf{F}^{\top}\_{\text{INV},i})) \\\\
> &\text{Apply R and t:}\\\\
> &=  \psi^{(k)} ( \boldsymbol{f}_i^{(k-1)}, \sum\_{j \in \mathcal{N}(i)} \phi^{(k)} (\boldsymbol{f}_i^{(k-1)}, \boldsymbol{f}_j^{(k-1)},||(p_iR-t)-(p_jR-t)||, ((p_iR-t)-(p_jR-t)) (\mathbf{F}\_{\text{global}}  R)^{\top}\mathbf{F}^{\top}\_{\text{INV},i})) \\\\
> &=  \psi^{(k)} ( \boldsymbol{f}_i^{(k-1)}, \sum\_{j \in \mathcal{N}(i)} \phi^{(k)} (\boldsymbol{f}_i^{(k-1)}, \boldsymbol{f}_j^{(k-1)},||p_i-p_j||, (p_i-p_j)RR^{\top} \mathbf{F}^{\top}\_{\text{global}} \mathbf{F}^{\top}\_{\text{INV},i})) \\\\
> &= \psi^{(k)} ( \boldsymbol{f}_i^{(k-1)}, \sum\_{j \in \mathcal{N}(i)} \phi^{(k)} (\boldsymbol{f}_i^{(k-1)}, \boldsymbol{f}_j^{(k-1)},e\_{ij}, \widehat{e}\_{ij} \mathbf{F}^{\top}\_{\text{global}} \mathbf{F}^{\top}\_{\text{INV},i})) \\\\
> \end{align}
>
> Applying a rotation $R$ and translation $t$ does not change the expression in Equation (5). Therefore, Equation (5) is unaffected by rotation and translation, indicating that it is SE(3)-invariant.
>
> We hope our response addresses your concerns.
>
>
> ****
>
> **$\bullet$ Response to weakness 4 and question 2**:
>
> We sincerely thank you for your valuable comments.
>
> We conducted experiments using iComFormer [1] as the backbone, which outperforms eComFormer. Below, we provided the evaluation of JARVIS dataset. Notably, our solution outperforms or at least is on par with the baselines. The preliminary results are as follows.
>
>
> ### Table: Property prediction results on the JARVIS dataset
>
> | Method                                               | Form. energy (eV/atom) | Total energy (eV/atom) | Bandgap (OPT) (eV) | Bandgap (MBJ) (eV) | E hull (eV) |
> |---------|--------|----------|-----|--------|--------------|
> | Crystalframer      | 0.0263    | 0.0279     | 0.117    | 0.242     | 0.047        |
> | eComFormer       | 0.0284      | 0.0315   | 0.124   | 0.283       | 0.044        |
> | — w/ SO(3)-equivariant Gram-Schmidt local frame      | 0.0285     | 0.0296     | 0.115| 0.271 | 0.043|
> | **— w/ Quaternion-based SPFrame (ours)**  | 0.0261  | 0.0276 | 0.107  | **0.239**           | **0.042**    |
> | iComFormer          | 0.0272     | 0.0288   | 0.122   | 0.261  | 0.047  |
> | — w/ SO(3)-equivariant Gram-Schmidt local frame      | runing    |runing       | 0.112     |  0.255     | 0.045        |
> | **— w/ Quaternion-based SPFrame (ours)**        | **0.0250**  | **0.0259** | **0.106** | 0.251               | **0.042**    |
>
> The performance improvement of SPFrame on another backbone (iComFormer) demonstrates its general effectiveness.
>
> We hope our response addresses your concerns.
>
>
>
> [1] Complete and efficient graph transformers for crystal material property prediction. *ICLR2024*.
>
>
>
>
> ****
>
> **$\bullet$ Response to weakness 3 and question 3**:
>
>
>
> We sincerely thank you for your comments. For fair comparisons, we just follow the experimental settings in the literature [1,2,3,4]. We agree that statistical results can be more rigorous. To reflect the statistical results, we evaluate various backbones for addressing the significance of our solution. Specifically, we replaced the backbone with a more powerful model to achieve a more significant improvement over the baseline. The detailed results are as iComFormer results (in Response to weakness 4 and question 2).
>
> We hope our response addresses your concerns.
>
>
> [1] Periodic graph transformers for crystal material property prediction. *NeurIPS2022*.
>
> [2] Complete and efficient graph transformers for crystal material property prediction. *ICLR2024*.
>
>
> [3] Crystalformer: Infinitely Connected Attention for Periodic Structure Encoding. *ICLR2024*.
>
> [4] Rethinking the role of frames for SE (3)-invariant crystal structure modeling. *ICLR2025*.
>
>
>
>
>
>
> ****
>
> **$\bullet$ Response to question 4**:
> Thank you for your comment. In fact, $\widehat{\boldsymbol{e}}_{ij}$ refers to the edge vector between atoms $i$ and $j$.
>
>
>
> ****
>
> **$\bullet$ Response to Typos**:
>
> We sincerely thank you for your careful review. We will revise these typos. We will remove the "rightarrow" notation on Line 98 and retain only the equation $(\mathbf{A},\mathbf{Q}_g\mathbf{X}+\mathbf{t}_g,\mathbf{Q}_g\mathbf{L})=(\mathbf{A},\mathbf{X},\mathbf{L})$.

---

> > ### Comment · Reviewer_drCP · 2025-08-05
> >
> > Thank you for taking the time to address my questions and comments. Most of my concerns have been resolved, and I have raised my score.
> >
> > However, I am curious why the authors chose to highlight eComFormer in the main text. In line 208, they state it was selected because it represents the state of the art, but this seems unjustified given that iComFormer shows stronger performance.

---

> > > ### Author Response · Authors · 2025-08-05
> > >
> > > Dear Reviewer drCP,
> > >
> > > Firstly, we sincerely thank you for your thoughtful comments and for raising the score.
> > >
> > > Below, we will address your question in detail.
> > >
> > > ****
> > > **$\bullet$ Response to the choice of highlighting eComFormer in the main text**:
> > >
> > > eComFormer and iComFormer are two methods proposed in the same paper [1], and overall, both models demonstrate competitive performance.
> > > Our proposed method is a general framework that improves performance across both eComFormer and iComFormer. This is also proved in our experimental results in the main text and the response in the rebuttal.
> > >
> > >
> > > **We used eComFormer in the main text because it allowed us to present our method more clearly and concisely.**
> > >
> > > Specifically, the message passing formulation of eComFormer can be directly expressed by Equation (2) in the main text (page 4):
> > >
> > > $\boldsymbol{f}_i^{(k)} =  \psi^{(k)} (\boldsymbol{f}_i^{(k-1)}, \sum _{j \in \mathcal{N}(i)} \phi^{(k)}(\boldsymbol{f}_i^{(k-1)}, \boldsymbol{f}_j^{(k-1)}, \boldsymbol{e} _{ij}, \widehat{\boldsymbol{e}} _{ij}))$
> > >
> > > This formulation provides a straightforward way to illustrate how our frame-based method is incorporated. When applying SPFrame, the message passing equation becomes:
> > >
> > >
> > > $\boldsymbol{f}_i^{(k)} = \psi^{(k)} ( \boldsymbol{f}_i^{(k-1)}, \sum _{j \in \mathcal{N}(i)} \phi^{(k)} ( \boldsymbol{f}_i^{(k-1)}, \boldsymbol{f}_j^{(k-1)},  \boldsymbol{e} _{ij}, \widehat{\boldsymbol{e}} _{ij}\mathbf{F}^{\top} _\text{global}\mathbf{F}^{\top} _{\text{INV},i}) )$
> > >
> > > On the other hand, iComFormer introduces additional complexity by incorporating the lattice matrix $L$ into its message passing formulation:
> > >
> > > $\boldsymbol{f}_i^{(k)} = \psi^{(k)} ( \boldsymbol{f}_i^{(k-1)}, \sum _{j \in \mathcal{N}(i)} \phi^{(k)} ( \boldsymbol{f}_i^{(k-1)}, \boldsymbol{f}_j^{(k-1)},  \boldsymbol{e} _{ij}, \xi(\widehat{ \boldsymbol{e}} _{ij},L)) )$
> > >
> > > where the function $\xi(\widehat{ \boldsymbol{e}} _{ij},L)$ computes the angles between the edge vector $\widehat{\boldsymbol{e}} _{ij}$ and the lattice basis vectors $l _{1,2,3} \in L$. When SPFrame is applied to iComFormer, the corresponding message passing becomes:
> > >
> > >
> > > $\boldsymbol{f}_i^{(k)} = \psi^{(k)} ( \boldsymbol{f}_i^{(k-1)}, \sum _{j \in \mathcal{N}(i)} \phi^{(k)} ( \boldsymbol{f}_i^{(k-1)}, \boldsymbol{f}_j^{(k-1)},  \boldsymbol{e} _{ij}, \xi(\widehat{ \boldsymbol{e}} _{ij} \mathbf{F}^{\top} _\text{global}\mathbf{F}^{\top} _{\text{INV},i},L\mathbf{F}^{\top} _\text{global})) )$
> > >
> > >
> > > As can be seen, the message passing formulation of iComFormer is more intricate, and the function $\xi( \cdot)$ requires additional explanation to fully convey the core idea of our method. This added complexity may detract from the clarity and conciseness of the main exposition.
> > >
> > >
> > > We are grateful for the reviewer's reminder. We will incorporate the results for both eComFormer and iComFormer into the revised manuscript and modify the sentence in Line 208 to eliminate potential ambiguity regarding our choice.
> > >
> > >
> > > We hope our response addresses your question.
> > >
> > > Best regards,
> > >
> > > The Authors
> > >
> > >
> > >
> > > [1] Complete and efficient graph transformers for crystal material property prediction. *ICLR2024*.

---

> > > > ### Comment · Reviewer_drCP · 2025-08-09
> > > >
> > > > Thank you for addressing my question. I have no further questions and support accepting the paper.

---

### Official Review · Reviewer_5W3r · 2025-07-03

**Clarity:** 3
**Significance:** 2
**Originality:** 3
**Rating:** 5
**Confidence:** 3

**Summary:**

Machine learning tools are used for predicting various properties of crystals. Many of these properties, such as potential energy of the system, are invariant with respect to rotations, and more generally, to rigid transformations of an input crystal. A naive approach to constructing a model that predicts invariant outputs is to limit the used features to only those that are invariant with respect to rotations; however, this limits the expressivity of the corresponding models. An alternative approach is to define global or local frames, which would canonicalize the crystal's orientation, thereby making the predictions naturally invariant with respect to rotations. Both purely global and local frames, at least one particular method used as a reference, have their own limitations; thus, the paper introduces a new method that combines both ideas to efficiently construct rotationally invariant architectures expressive enough for the accurate prediction of potential energy and other properties of crystals.

**Questions:**

Is your method smooth or not?

Did I understand correctly that the features f in equations 2 and 3 are invariant with respect to rotations, and not equivariant ones such as those of Tensor Field Network, Nequip, Mace, and many others?

Do you think that your method is more or less efficient compared to equivariant Graph Neural Networks, which are the de facto standard in the field, and for which there exists the mentioned completeness theorem (along with a few others)?

**Ethical Concerns:**

["NO or VERY MINOR ethics concerns only"]

**Final Justification:**

I appreciate the clarifications made by the authors, and I am willing to increase my score.

**Limitations:**

Some of the limitations are addressed in the text.

The paper would benefit, in my opinion, from the discussion of such issues as the lack of a formal universal approximation theorem to back up the claim of superior expressivity, and potential questions of smoothness.

**Paper Formatting Concerns:**

I have not noticed paper formatting concerns.

**Quality:**

4

**Strengths And Weaknesses:**

The paper clearly identifies limitations of the previous work and proposes a way to overcome these issues. Additionally, it provides thorough benchmarks on more than one dataset against multitude of other state-of-the-art models. On top of that, it provides a clear analysis of the required computational cost.

However, I have a few concerns regarding this paper. The first one is related to the lack of any sort of formal and systematic analysis of the expressive power, which the paper claims to improve compared to previous approaches. For many machine learning architectures, there exist explicit completeness theorems that establish that these machine learning architectures are universal approximators. There is, for instance, universal approximation theorem for MLPs, Kolmogorov Arnold representation theorem for KANs, and, the most relevant for machine learning on crystal structures, the one which states that certain class of rotationally equivariant graph neural networks is a universal approximator when predicting invariant properties for crystal structures, the one introduced here "On the Universality of Rotation Equivariant Point Cloud Networks" (I am not affiliated). The reviewed paper claims that it leads to increased expressivity of the corresponding machine learning models, but doesn't provide any formal definition of this increase. It is unclear whether the proposed method leads to architectures featuring a universal approximation property, similarly to the aforementioned approaches, or not. In the current way of representation, it is not even 100% convincing that the proposed method is strictly more expressive compared to the discussed approach based solely on local frames. It is well illustrated that the proposed method eliminates some of the issues of solely local frames, but it can simultaneously introduce new ones.

My another concern is the smoothness of the method with respect to geometric deformations of input crystal configurations. Can it be that a small movement of the atoms introduces rapid and discontinuous change in the predicted potential energy or other property? This property can be as important, if not even more, than the rotational invariance to run practical molecular dynamics simulations. While not 100% sure, I got the impression that the proposed method can sometimes introduce discontinuous jumps of local and global frames, which in turn would lead to discontinuities in the predicted target property. In the case of Gram-Schmidt diagonalization, this can occur when the initial vectors are collinear. For quaternions, the problematic case potentially leading to discontinuities is when a = b = c = d = 0. It would be helpful if the authors could clarify this matter.

---

> ### Author Rebuttal · Authors · 2025-07-28
>
> We sincerely thank you for your constructive suggestions. Below we will address your questions in detail.
>
> ****
>
> **$\bullet$ Response to weakness 1 (formal analysis of the expressive power)**:
>
> We sincerely thank you for your constructive suggestions.
>
> We provide a mutual information-based proof to explain why the SPFrame approach yields more informative node/atom representations than the local frame method.
>
> Let a crystal structure be represented as a graph with $N$ atoms. Denote the node/atom feature corresponding to atom $i$ as $f_i$.  We define the complete set of node/atom features under two different framing schemes as follows:
>
> $$A=\\{f_{local,i}|i=1,2,..,N\\}, B=\\{f_{sp,i}|i=1,2,..,N\\},$$
>
> where $A$ corresponds to node/atom features in the network using local frames, and $B$ corresponds to those in the network using the SPFrame approach.
>
> We assume that the atoms with indices $p,q$ ($1<p,q<N$) are symmetry-equivalent.
> The use of the local frame results in identical local environments for atoms p and q (According to the illustration in Figure 2 in our paper, page 4). As a consequence, after message passing via Equation (3) (in our paper, page 4), we can have
>
> $$f_{local,p}=f_{local,q}. $$
>
> Consequently, the effective number of distinguishable node/atom representations is reduced. Denoting the cardinality as $|\cdot|$, we can have:
>
> $$|A|=N-1<N.$$
>
> In contrast, the SPFrame approach preserves the differences in the local environments of atoms p and q. After message passing via Equation (5) (in our paper, page 5), the node/atom representations satisfy:
>
> $$f_{local,p} \neq f_{local,p}  \implies |B|=N.$$
>
>
> For scalar crystal property prediction, the final node features are first aggregated to get a global graph-level representation. This graph-level representation is then passed through a regression head. We denote the prediction target as a variable  $Y \in \mathcal{Y}$. The neural network induces the following mapping:
>
>
> $$h_{local}:A \mapsto \mathcal{Y}, \quad h_{sp}:B \mapsto \mathcal{Y}.$$
>
> Since $|B|>|A|$, there exists a surjective mapping $g_1:B \mapsto A$, such that $h_{local}\circ g_1=h_{sp}$. However, an injective mapping $g_2:A \mapsto B$ does not exist in general due to loss of distinguishability in $|A|$. Consequently, the information flow can be described via the following Markov chain:
> $$Y \to B \to A.$$
>
> Applying the data processing inequality to this chain yields:
> $$I(Y;B) \ge I(Y;A),  $$
>
> with equality if and only if:
>
> $$I(B;Y | A) =0 \quad\text{and}\quad Y \to A \to B$$
>
> i.e., the chain $Y \to A \to B$ is also valid. However, the absence of a mapping $g_2:A \mapsto B$ implies that this reverse chain cannot be constructed. Therefore, the inequality is strict:
> $$I(Y;B) \gt I(Y;A),  $$
>
>
> This result implies that the node/atom representations obtained via SPFrame retain strictly higher mutual information with the target variable $Y$ than those obtained via the local frame method. In other words, SPFrame-based features preserve more task-relevant information.
>
> We hope our response can address your concerns.
>
>
>
> ****
>
> **$\bullet$ Response to weakness 2 (smooth) and questions 1**:  We sincerely thank you for your insightful comments. In practical applications, our method can be made smooth. Take the Quaternion-based SPFrame as an example. In our implementation, for a quaternion (a,b,c,d),  we only predict the values of b,c, and d, while a is fixed to 1 by default (see our Anonymous GitHub repository, file ./models/transformer.py,  line 229, function quaternion_1ijk_to_rotation_matrix(q)). This design prevents the network from encountering the discontinuity issue when a = b = c = d = 0 during the normalization step in rotation matrix construction.
>
> Moreover, this paper focuses primarily on scalar property prediction for crystals, which does not impose smoothness requirements. If one aims to extend the method to tasks such as force prediction, smooth rotation matrix construction methods can be adopted to inherently avoid discontinuities, such as the SVD-based rotation projection proposed in [1].
>
> We hope our response addresses your concerns.
>
>
> [1] Learning with 3D rotations, a hitchhiker’s guide to SO(3). *ICML2024*.
>
> ****
>
> **$\bullet$ Response to questions 2**:
>
> Thank you for your question. This work primarily investigates the application of frame-based methods in the prediction of scalar properties of crystals, where invariance is required. Therefore, the features f in equations 2 and 3 are invariant with respect to rotations. In contrast, methods such as NequIP focus on predicting interatomic potentials, where equivariance is necessary for accurate force prediction. As a result, these methods incorporate spherical harmonics and tensor products in their network architecture to construct equivariant representations. We hope our response addresses the reviewer’s question.
>
> ****
>
> **$\bullet$ Response to questions 3**:
>
> We sincerely thank you for your comments.
> Frame-based methods have already been applied to force prediction tasks [1,2], as they decouple the network from the strict requirement of equivariance. This allows them to avoid the high computational costs associated with architectures that rely on spherical harmonics and tensor products to achieve equivariance. Instead, frame construction methods offer lower computational overhead compared to tensor products, enabling more efficient models. This work primarily focuses on the application of frame-based methods for scalar property prediction in crystals, where the goal is to achieve invariance and to improve predictive accuracy. If our method is extended to force prediction tasks, it can theoretically achieve higher efficiency in the same manner as shown in [1,2].
>
>
> [1] Graph neural network with local frame for molecular potential energy surface. *LoG2022*.
>
> [2] A new perspective on building efficient and expressive 3D equivariant graph neural networks. *NeurIPS2023*.

---

> > ### Comment · Reviewer_5W3r · 2025-08-08
> > **Response**
> >
> > I appreciate the clarifications made by the authors, and I am willing to increase my score.

---

> > > ### Author Response · Authors · 2025-08-09
> > >
> > > Dear Reviewer 5W3r,
> > >
> > > We sincerely thank you for your valuable comments and for raising the score.
> > >
> > > Best regards,
> > >
> > > The Authors

---

### Decision · Program_Chairs · 2025-09-17

**Decision:**

Accept (poster)

**Comment:**

The paper introduces a methodology for crystal property prediction based on local and global frames. The proposed methodology constructs invariant local frames as well as an equivariant global frame to ensure SO(3) invariance. The paper has merits that have been identified by the reviewers, including the importance of the problem addressed and the fact that SPFrame can be easily integrated with existing models. During the discussion period, the authors provided clarifications and addressed several of the reviewers' concerns. Overall, the contribution is well motivated, practically useful, and of interest to the community of crystal structure modeling.